# Highly oxidized products from the atmospheric reaction of hydroxyl radicals with isoprene

Torsten Berndt ✉, Erik H. Hoffmann, Andreas Tilgner & Hartmut Herrmann

Isoprene ($C_5H_8$) globally accounts for half of the non-methane hydrocarbon flux into Earth´s atmosphere. Its degradation is mainly initiated by the gas-phase reaction with OH radicals yielding a complex system of $RO_2$ radicals. Subsequent product formation is not conclusively understood yet. Here we report the observation of $C_4$- and $C_5$-products from OH + isoprene bearing at least two functional groups. Their production is initiated either by the reaction of initially formed δ-$RO_2$ radicals with NO or by 1,6 H-shift isomerization of Z-δ-$RO_2$ radicals. Both reaction channels also form highly oxygenated molecules (HOMs), which could be important for the generation of secondary organic aerosol. $C_5H_9O_8$ and $C_5H_9O_9$ radicals represent the main precursors of closed-shell HOMs. Global simulations revealed that the isoprene-derived HOM-$RO_2$ production is comparable with that of α-pinene, currently regarded as very important HOM source. This study provides a more complete insight into isoprene´s degradation process including the HOM formation.

Biogenic emissions are responsible for about 90 % of the non-methane hydrocarbon flux into Earth´s atmosphere[1]. Isoprene is regarded as the most important non-methane compound in this process with an annual emission rate of about 600 million metric tons of carbon[2]. Its dominant atmospheric sink is the gas-phase reaction with OH radicals, which leads to production of a variety of $RO_2$ radicals due to isoprene´s conjugated diene structure. Detailed knowledge of the $RO_2$ chemistry is needed for the understanding of subsequent product channels and the occurring $HO_x$ (OH and $HO_2$ radicals) regeneration[3]. The latter can perceptibly influence the whole atmospheric oxidation system due to isoprene´s huge emission rate.

According to the currently available mechanistic insights, the OH attack towards isoprene predominantly proceeds by addition at the terminal positions, 63 % in 1-position and 37 % in 4-position, forming E- and Z-OH-adducts, i.e., **2** and **3**, respectively, from the OH reaction in 1-position (Fig. 1)[4]. Subsequent $O_2$ addition leads to the corresponding δ- and β-$RO_2$ radicals **4**−**6** with an initial relative abundance **4** / **5** / **6** of 14 % / 79 % / 6 %[5]. OH addition in 4-position yields the analogous $RO_2$ radicals, **4´**, **5´** and **6´**, with a similar initial distribution of 14 % / 70 % / 16 %[5]. Distinct reversibility of the $O_2$

addition in allylic systems[6] allows efficient $RO_2$ interconversion. About 10−60 s of time are required in order to get an equilibrated $RO_2$ radical distribution[5,7] which is featured by a higher fraction of thermodynamically preferred β-$RO_2$ radicals **5** (**5´**) of > 90 %[5,8]. The $RO_2$ radicals react further either bimolecularly with NO and $HO_2$ radicals mainly or via $RO_2$ isomerization in the case of the Z-δ-$RO_2$ radical **6** (**6´**) forming the allyl species **10** (**10´**) via 1,6 H-shift[3]. In the following, it will not be mentioned explicitly that the corresponding $RO_2$ radicals and other intermediates from the OH attack in 1- and 4-position undergo analogous reactions.

In the reaction of the HO-$C_5H_8O_2$ radicals **4**, **5** and **6** with NO (Fig. 1), organic nitrates, $RONO_2$, are formed with an reported average molar yield of 13 ± 4 %[5] besides the corresponding alkoxy radicals[9]. **4** + NO leads to the E-δ-alkoxy radical **7** that very rapidly isomerizes into the Z-δ form **9** via an epoxy intermediate **8**[10]. Thus, both δ-$RO_2$ radicals **4** and **6** end up in the alkoxy species **9** that further isomerize via 1,5 H-shift forming the allyl radical **11**[11]. The final main product from **5** + NO is methyl vinyl ketone (MVK) and formaldehyde, being along with methacrolein (MACR) and formaldehyde from **5´** + NO the most important products from isoprene oxidation in the presence of NO[4,8].

Atmospheric Chemistry Department (ACD), Leibniz Institute for Tropospheric Research (TROPOS), 04318 Leipzig, Germany. ✉e-mail: berndt@tropos.de

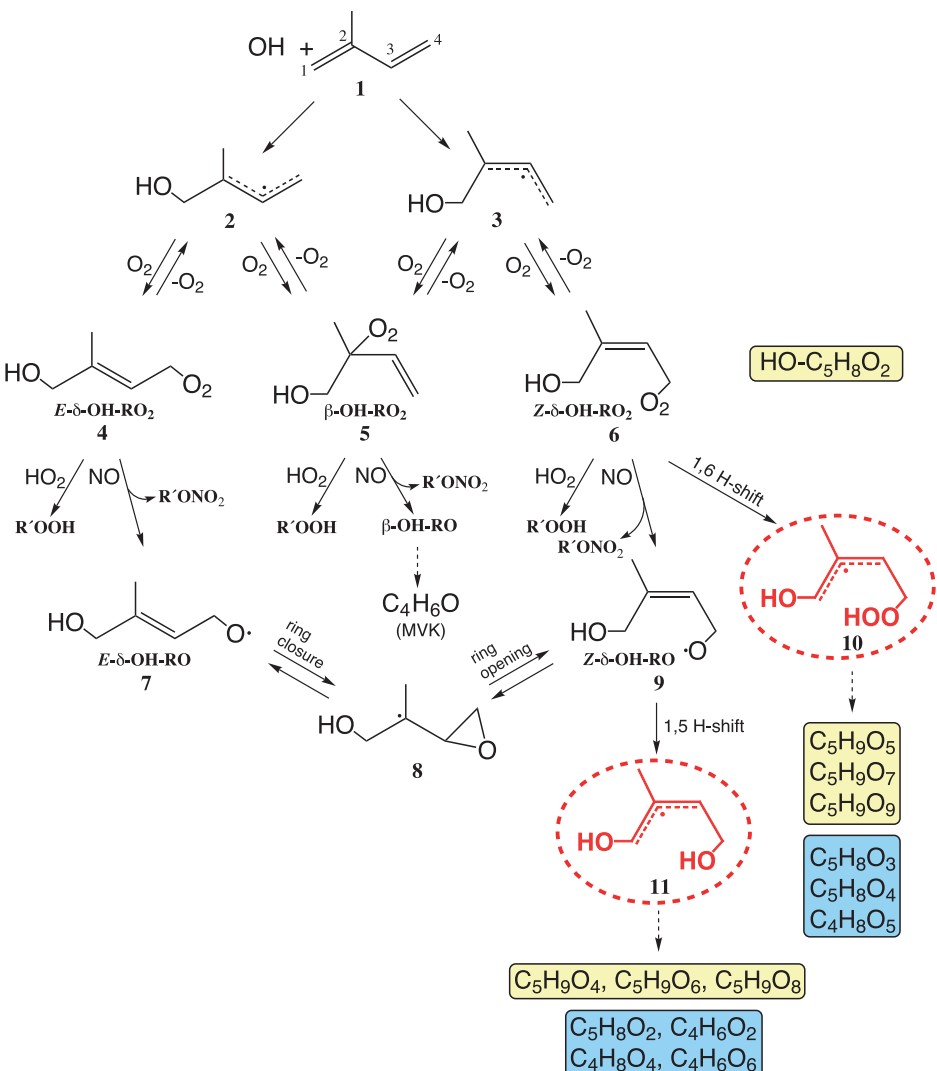

**Fig. 1 | First reaction steps of the OH radical-initiated oxidation of isoprene for the OH attack in 1-position.** Product formation from the OH attack in 4-position proceeds in an analogous way[4]. Signals of detected RO$_2$ radicals are highlighted in yellow and those from closed-shell products in blue. Only main products of individual pathways are shown. Dashed arrows indicate complex reactions to the products.

The main products of the HO-C$_5$H$_8$O$_2$ radical reactions with HO$_2$ are the corresponding hydroxy hydroperoxides, ROOHs[4,9]. Reported molar ROOH yields are 93.7 ± 2.1 %[12] and 88 ± 12 %[13] qualifying the HO$_2$ reaction as an important radical termination step. Other RO$_2$ radical paths, such as possible RO$_2$ self- and cross-reactions[4,9,14] or the reaction with OH radicals forming hydrotrioxides[15], are neglected here due to their small importance for the RO$_2$ radical balance under atmospheric conditions.

The allyl radicals **10** and **11** are structurally very similar differing only by the OOH and OH group (Fig. 1). Thus, a similar chemical behavior and similar product channels can be expected in their further reactions.

Subsequent reactions of **10** were already subject of a series of theoretical[3,7,16–18] and experimental investigations[5,14,17,19–22] reporting a distinct autoxidation process[23,24] that leads to RO$_2$ radicals with up to nine O atoms (Fig. 2). While hydroperoxy aldehydes (HPALDs, C$_5$H$_8$O$_3$) **15** were analyzed at first as the only closed-shell products[19,20], more recent studies disclosed a more complex product distribution[5,14,17]. HPALDs **15** and dihydroperoxy carbonyls (C$_4$H$_8$O$_5$) **20** are currently considered as the main 1,6 H-shift products in present atmospheric models[21,25]. In the experiments, however, C$_4$H$_8$O$_5$ has been detected up to now only in traces[14].

For the product formation starting from the allyl radical **11**, reversible O$_2$ addition is expected that leads to the corresponding α- and γ-RO$_2$ radicals **21**–**23** (Fig. 3)[4,5,17]. The α-hydroxy RO$_2$ radical **21** can readily release HO$_2$[26] forming the unsaturated hydroxy carbonyl **24**, which has been observed in experiments in the presence of NO[5,27,28]. A next isomerization step via 1,6 H-shift is possible for the Z-form of the γ-RO$_2$ radical **22**[4,5,17] producing the higher oxidized RO$_2$ radical **26** after O$_2$ addition. It has been proposed that **26** can either form a next, higher oxidized RO$_2$ radical **27** after H-shifts or a hydroxy-hydroperoxy carbonyl **28** accompanied by OH release[17]. A signal consistent with the formation of the highest oxidized RO$_2$ radical **27** was already observed under conditions of significant RO$_2$ self- and cross-reactions in experiments where the allyl radical **11** can be produced via the alkoxy channels of the reactions between **4** or **6** with other RO$_2$ radicals[17,22].

Here we experimentally show the formation of C$_4$- and C$_5$-products with at least two functional groups from the OH + isoprene reaction for near-atmospheric conditions including the NO level. Particular attention is paid to the product formation initiated by δ-RO$_2$ + NO reactions and to the generation of highly oxygenated molecules (HOMs)[24], that are considered as possible precursors of the secondary organic aerosol (SOA). Accompanied modeling on the

**Fig. 2 | Proposed reaction scheme for consecutive reactions of the allyl radical 10 from the 1,6 H-shift channel.** The scheme is consistent with the observed product formation from the present study as well as partly with data from the literature[3–5,7,14,17]. Signals of detected RO$_2$ radicals are highlighted in yellow and those from closed-shell products in blue. Only important main products of individual pathways are shown. Bimolecular reactions of RO$_2$ radicals with NO and HO$_2$ radicals are not depicted.

global scale indicates the importance of the findings within isoprene's oxidation system.

## Results and discussion

The experiments were conducted at 295 ± 0.5 K under atmospheric conditions in a laminar flow tube (LFT)[29,30] with a reaction time of 32 s, thus, coming close to conditions of the equilibrated initial RO$_2$ radical distribution as existing in the atmosphere. Product analysis was carried out by means of a mass spectrometer with a detection limit down to ~$10^4$ molecules cm$^{-3}$ applying a suite of reagent ions, i.e., iodide (I$^-$), ethylaminium (C$_2$H$_5$NH$_3^+$) and nitrate (NO$_3^-$)[14]. OH radicals were generated either by photolysis of H$_2$O$_2$ or isopropyl nitrite (IPN) or by ozonolysis of tetramethylethylene (TME). NO concentrations ranged from < 2 × $10^8$ to 8.3 × $10^{10}$ molecules cm$^{-3}$ which are relevant for remote to urban areas. The OH + isoprene reaction rate was in the range of (2.7–28) × $10^6$ molecules cm$^{-3}$ s$^{-1}$ for low-NO conditions covering the atmospheric rate of ~(5–10) × $10^6$ molecules cm$^{-3}$ s$^{-1}$, [OH] = 1 × $10^6$ and [C$_5$H$_8$] = (5–10) × $10^{10}$ molecules cm$^{-3}$. Thus, concentrations of initially formed RO$_2$ radicals, HO-C$_5$H$_8$O$_2$ (Fig. 1), could not be higher than their atmospheric levels making subsequent product formation observed in these experiments applicable to atmospheric conditions. Results from modeling of the initial RO$_2$ radical processes[4,7] using latest kinetic

data[4,5] allowed to assess the importance of the 1,6 H-shift and the δ-RO$_2$ + NO channel for product formation in the experiments, see Supplementary Table 1.

### Detectable products for background NO level

The observed product formation in the LFT for background [NO] < 2 × $10^8$ molecules cm$^{-3}$ confirmed the findings of our previous study with shorter reaction times of 3–7.9 s[14] and revealed some new insights. Recorded spectra using OH generation either via TME ozonolysis or H$_2$O$_2$ photolysis were in good agreement (Supplementary Fig. 1). Besides the known signals attributed to the RO$_2$ radicals, HO-C$_5$H$_8$O$_2$ **4–6**, C$_5$H$_9$O$_5$ **12–14** and C$_5$H$_9$O$_7$ **18**, and the closed-shell products, C$_5$H$_8$O$_3$ **15**, C$_5$H$_8$O$_4$ **17** and C$_4$H$_8$O$_5$ **20**[14], the occurrence of C$_5$H$_8$O$_2$ compounds became clearly visible. Reanalysis of old data sets[14] revealed that such signals were also present there, but unfortunately overlooked in the former analysis. Figure 4 shows observed concentrations of 1,6 H-shift products formed via the allyl radical **10** along with C$_5$H$_8$O$_2$ from a measurement series applying iodide ionization. Needed calibration factors were obtained from a convergence method using a series of ionization schemes, see Methods. The RO$_2$ radical concentrations featured a linear increase with rising isoprene

**Fig. 3 | Proposed reaction scheme for consecutive reactions of the allyl radical 11 from the δ-RO₂ + NO channel.** The scheme is consistent with the observed product formation from the present study as well as with suggestions from the literature[4,5,11,17,27,28]. Signals of detected RO₂ radicals are highlighted in yellow and those from closed-shell products in blue. Only main products of individual pathways are shown. Bimolecular reactions of RO₂ radicals with NO and HO₂ radicals are not depicted.

conversion, which confirmed the absence of significant RO₂ self- and cross-reactions under the chosen conditions (Supplementary Fig. 2).

$C_5H_8O_2$ concentrations, very similar to those of the HPALDs **15**, linearly increased with rising isoprene conversion in an almost identical way to all 1,6 H-shift products. $C_5H_8O_2$'s structure and possible formation routes, however, are speculative at the moment. Reactions of δ-RO₂ radicals **4** and **6** with background NO can only partly explain the $C_5H_8O_2$ production if formation of the hydroxy carbonyl **24** is assumed. Moreover, significant contributions from HO-$C_5H_8O_2$ radical self- and cross-reactions are unlikely, because of the 2nd order kinetics of this process, which is contrary to the observed linear increase of $C_5H_8O_2$ with rising isoprene conversion.

The molar yields of 1,6 H-shift products (regarding reacted $C_5H_8$) measured in the LFT are similar or slightly higher compared to the previous results obtained in the free-jet flow system for a reaction time of 7.9 s[14]. Rising importance of 1,6 H-shift isomerization of **6** with time can be expected for sufficiently low NO and HO₂ radical levels according to the current mechanistic and kinetic description of the HO-$C_5H_8O_2$ radical system (Fig. 1)[3–5,7], which would justify raising product yields. The molar HPALD yield is also in reasonable agreement with the result of an earlier study performed in the LFT reporting a

HPALD yield of $4^{+4}_{-2}$ % regarding reacted isoprene[20]. Furthermore, the formation of $C_4H_8O_5$ **20**, the expected decomposition product of $C_5H_9O_7$ radicals **18** (Fig. 2), was only a minor process (Fig. 4) being in line with our former findings[14]. This is in contradiction to conclusions based on high-temperature OH radical recycling experiments (480–584 K) stating the formation and subsequent decomposition of $C_5H_9O_7$ **18** as the main process of 1,6 H-shift product formation for atmospheric conditions[31].

Product analysis by means of ethylaminium as the reagent ion additionally disclosed the formation of $C_5H_9O_9$ radicals **19** (Fig. 2) not detected for shorter reaction times in our previous study[14]. This finding was further confirmed by measurements applying nitrate ionization, frequently used to selectively measure HOMs[24], that already earlier revealed $C_5H_9O_9$ production in other experiments carried out at higher concentration levels[17,22]. The HOM-RO₂ radical concentrations of $C_5H_9O_7$ **18** and $C_5H_9O_9$ **19** determined in the present study by different ionization schemes are in good agreement (Supplementary Fig. 3), supporting the reliability of stated concentrations.

**Effect of NO addition**

Even small NO additions of $(3.2–32) × 10^8$ molecules cm⁻³ resulted in a measurable decline of 1,6 H-shift product formation caused by the

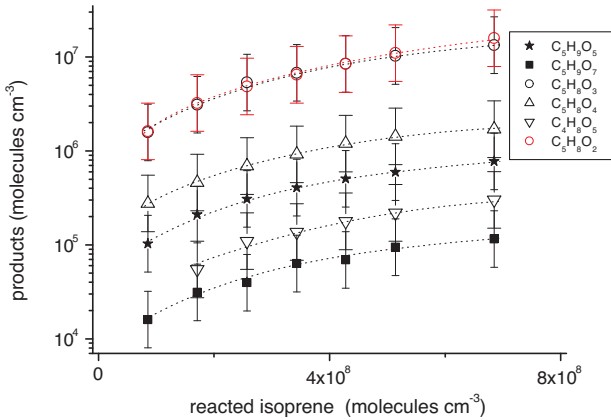

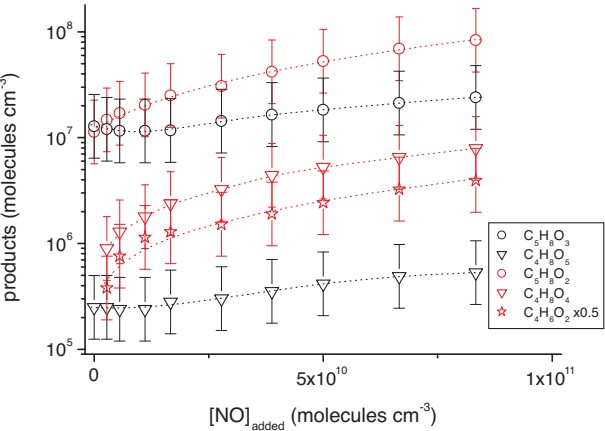

**Fig. 4 | Concentrations of 1,6 H-shift products and $C_5H_8O_2$ as a function of converted isoprene for background NO conditions.** $RO_2$ radicals are depicted with full and closed-shell products with open symbols. OH radicals were generated via TME (tetramethylethylene) ozonolysis and iodide served as the reagent ion. The amount of reacted isoprene was calculated based on a comprehensive reaction mechanism, see Methods. $C_5H_8O_2$ concentrations are lower limit values. The error bars represent the uncertainty of a factor of ~2. Reactant concentrations were [TME] = $(1.0–8.0) \times 10^{10}$, [isoprene] = $(1.25–10) \times 10^{11}$ and [$O_3$] = $3.0 \times 10^{11}$ molecules cm⁻³.

**Fig. 5 | Closed-shell product concentrations as a function of added NO.** Products formed via the 1,6 H-shift channel are given with black symbols and those from the δ-$RO_2$ + NO channel with red symbols. $C_5H_8O_4$ analysis was influenced by changing background in the course of the measurement and is not presented here. TME (tetramethylethylene) ozonolysis served as the OH radical source and product analysis was carried out by iodide ionization. The error bars show the uncertainty of a factor of ~2. $C_5H_8O_2$ and $C_4H_6O_2$ concentrations are lower limit values. Reactant concentrations were [TME] = $8.0 \times 10^{10}$, [isoprene] = $1.0 \times 10^{12}$, [$O_3$] = $3.0 \times 10^{11}$ molecules cm⁻³ and added NO was in the range of $(2.8–83) \times 10^9$ molecules cm⁻³.

HO-$C_5H_8O_2$ + NO reaction lowering the $Z$-δ-$RO_2$ radical level in the system (Fig. 1 and Supplementary Fig. 4). At the same time, the signals of $C_5H_8O_2$, most likely due to the formation of the hydroxy carbonyl **24**, and the organic nitrates HO-$C_5H_8ONO_2$ distinctly increased. The measured onset of organic nitrate detection around $3 \times 10^8$ molecules cm⁻³ confirmed the stated background NO level < $2 \times 10^8$ molecules cm⁻³ (Supplementary Fig. 4).

A more complete insight into the NO-mediated $C_4$- and $C_5$-product formation, other than methyl vinyl ketone (MVK) and methacrolein (MACR)[4,8], is visible for higher NO concentrations of up to 8.3 × 10¹⁰ molecules cm⁻³. Using TME ozonolysis for OH generation, the reaction flux over the 1,6 H-shift channel initially declined with rising NO, but increased afterwards due to rising importance of the NO + $HO_2$ → OH + $NO_2$ reaction leading to rising isoprene conversion, see results from modeling in Supplementary Fig. 5. On the other hand, product formation via the δ-$RO_2$ + NO channel continuously increased with rising NO. This behavior is also visible in the product traces and allowed a rough assignment, i.e., $C_5H_8O_3$ **15** and $C_4H_8O_5$ **20** mainly formed from the 1,6 H-shift channel (Fig. 2) and $C_5H_8O_2$ **24**, $C_4H_8O_4$ **28** and $C_4H_6O_2$ from the δ-$RO_2$ + NO channel (Fig. 3), see Fig. 5. $C_4H_6O_2$ formation can be explained by the reaction of the $RO_2$ radicals **22** and **23** with NO and subsequent scission of the resulting β-hydroxy alkoxy radicals[32]. It is to be noted, that the measurements indicated the possible formation of $C_4H_6O_4$ as well, which could be formed from the $C_5H_9O_6$ radicals **26** in a similar way as for $C_4H_6O_2$. However, a reliable data analysis for $C_4H_6O_4$ was impossible due to the presence of $C_5H_{10}O_3$ (most likely the hydroperoxide HO-$C_5H_8OOH$) with a very similar mass, only poorly separable with our mass spectrometer. More generally, product assignment to a certain channel was not fully clear in each case. For instance, $C_4H_6O_2$ could partly also arise from the corresponding β-scission of alkoxy radicals formed from the $RO_2$ radicals **13** and **14**.

Organic nitrate formation via $RO_2$ + NO → $RONO_2$, or for the OH + isoprene system $C_5H_9O_x$ + NO → $C_5H_9O_{x-1}NO_2$, was well detectable for the HO-$C_5H_8O_2$ radicals as well as for the higher oxidized radicals $C_5H_9O_{4,5,6,7}$ applying iodide ionization (Supplementary Fig. 6). This finding supports the occurrence of the proposed $RO_2$ radicals according to the autoxidation process as given in Fig. 2 (1,6 H-shift

channel) and Fig. 3 (δ-$RO_2$ + NO channel). The organic nitrates arising from $C_5H_9O_8$ **27** and $C_5H_9O_9$ **19** were followed more conveniently by means of nitrate ionization, see later on. It is to be noted, that formed alkoxy radicals from $RO_2$ + NO can undergo internal H-shifts from OOH groups (if present) forming new R´$O_2$ radicals with one O atom less as experimentally observed in other reaction systems[33,34]. For instance, the reaction of $C_5H_9O_7$ **18** with NO could lead to the R´$O_2$ radical $C_5H_9O_6$´ not distinguishable from the structurally different $C_5H_9O_6$ **26** by means of mass spectrometry. Thus, the observed $RO_2$ radicals and subsequent formation of organic nitrates etc. from the 1,6 H-shift channel and the δ-$RO_2$ + NO channel could partly be merged.

The experiments were repeated utilizing IPN photolysis as another OH radical source where the OH production proceeds via NO + $HO_2$ → OH + $NO_2$. OH radical concentrations were measured indirectly by adding $SO_2$ to the reaction gas, not disturbing the OH + isoprene reaction, and following the $SO_3$ production from OH + $SO_2$, see Methods. The concentrations of converted isoprene deduced from the experimental OH data were in good agreement with the results from modeling, that inspires more confidence in the calculated reaction flux over both channels by the model (Supplementary Fig. 7). All product traces behaved as expected from the measurements before (Fig. 5), i.e., $C_5H_8O_2$ **24**, $C_4H_8O_4$ **28** and $C_4H_6O_2$ followed the δ-$RO_2$ + NO channel, etc., supporting their assignment (Supplementary Fig. 8).

The channel-specific molar product yields from the measurement series with different OH sources were in good agreement within the experimental uncertainty. Hence, molar yields of ~47 %, ~5.5 % and ~4.5 % for $C_5H_8O_2$, $C_4H_8O_4$ and $C_4H_6O_2$, respectively, can be derived for the δ-$RO_2$ + NO channel with an uncertainty of a factor of ~2 (Supplementary Fig. 9). The yields of $C_5H_8O_2$ and $C_4H_6O_2$ are lower limit values. Taking into account an organic nitrate yield of 13 ± 4 % from HO-$C_5H_8O_2$ + NO[5], all products detected amount to ~70 % on molar scale (uncertainty: factor of ~2). The stated $C_5H_8O_2$ yield is in very good agreement with results from a chamber study reporting $C_5$ hydroxy carbonyl yields of 45 ± 10 % from the OH attack in 1- and 4-position each[5]. Formation yields with respect to converted isoprene of 19.3 ± 6.1 % and 3.3 ± 1.6 % for $C_5H_8O_2$ and $C_4H_6O_2$, respectively[27], were found in a flow tube study using elevated concentration levels,

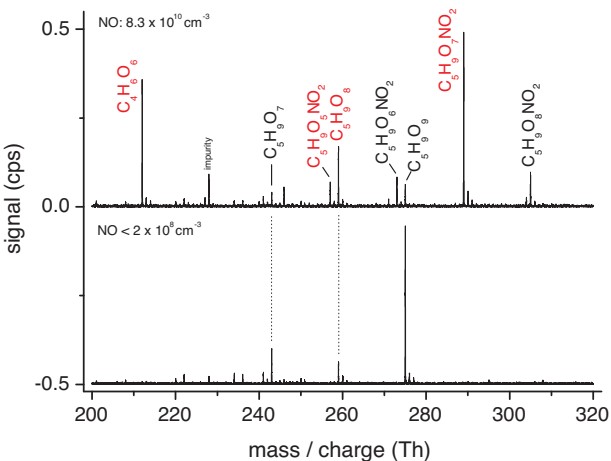

**Fig. 6 | Mass spectra in the HOM range from the OH + isoprene reaction for different NO levels.** (HOM: highly oxygenated molecule) Products assigned to the 1,6 H-shift channel are indicated in black and those from the δ-RO$_2$ + NO channel in red. TME (tetramethylethylene) ozonolysis served as the OH radical source and product analysis was carried out by nitrate ionization. Products appeared as the adduct with nitrate, i.e., their masses are shifted by 61.99 Th. The spectrum for the background NO level < 2 × 10$^8$ molecules cm$^{-3}$ is vertically moved by -0.5 units. Reactant concentrations were [TME] = 8.0 × 10$^{10}$, [isoprene] = 1.0 × 10$^{12}$, [O$_3$] = 3.0 × 10$^{11}$ molecules cm$^{-3}$ and added NO was either absent (lower part) or 8.3 × 10$^{10}$ molecules cm$^{-3}$ (upper part).

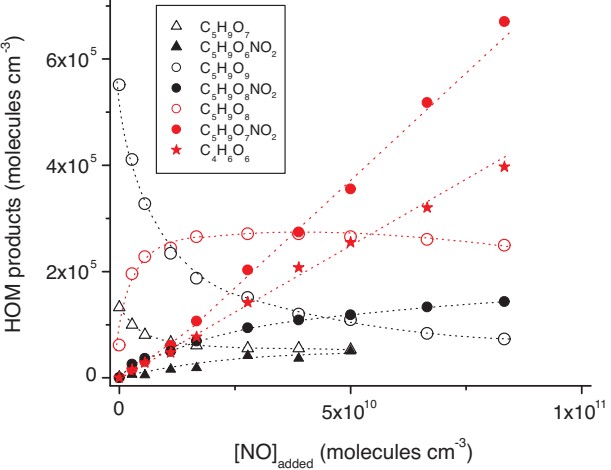

**Fig. 7 | Important HOM products as a function of added NO.** (HOM: highly oxygenated molecule) Products from the 1,6 H-shift channel are given in black and those from the δ-RO$_2$ + NO channel in red. OH radicals were produced from TME (tetramethylethylene) ozonolysis and the product analysis was carried out by nitrate ionization. Concentrations of C$_5$H$_9$O$_7$ and C$_5$H$_9$O$_6$NO$_2$ are omitted for highest NO additions because of possible signal overlapping. The uncertainty of product concentrations was a factor of ~2. Error bars are not shown for better clearness. Reactant concentrations were [TME] = 8.0 × 10$^{10}$, [isoprene] = 1.0 × 10$^{12}$, [O$_3$] = 3.0 × 10$^{11}$ molecules cm$^{-3}$ and added NO was in the range of (2.8–83) × 10$^9$ molecules cm$^{-3}$.

which stands for higher δ-RO$_2$ + NO channel-specific yields of both products than observed in the present study (~100 % in the case of C$_5$H$_8$O$_2$).

## HOM formation depending on the NO concentration

Nitrate ionization for product analysis was chosen in the experiments for selective and sensitive detection of HOMs, which should contain by definition at least six O atoms in the molecule[24]. Fig. 6 shows spectra recorded for the background NO level < 2 × 10$^8$ and the highest NO addition of 8.3 × 10$^{10}$ molecules cm$^{-3}$ using OH radical generation from TME ozonolysis. The RO$_2$ radicals C$_5$H$_9$O$_7$ **18** and C$_5$H$_9$O$_9$ **19** from the 1,6 H-shift channel (Fig. 2) were almost exclusively observed in the absence of NO addition, see lower part in Fig. 6. In the presence of NO additions, the product distribution became more complex due to additional HOM-RO$_2$ radical formation from the δ-RO$_2$ + NO channel, i.e., C$_5$H$_9$O$_6$ **26** and C$_5$H$_9$O$_8$ **27**, and the reaction products of all HOM-RO$_2$ radicals with NO, see upper part in Fig. 6. Detected main closed-shell HOM products are the four possible organic nitrates, C$_5$H$_9$O$_{5,6,7,8}$NO$_2$, and a C$_4$H$_6$O$_6$ compound. The latter can be formed from the C$_5$H$_9$O$_8$ radical **27** after 1,7 H-shift from the OOH group[35,36] and subsequent NO reaction forming the alkoxy radical in β-position to the OH moiety, that readily decomposes[32] finally leads to C$_4$H$_6$O$_6$ formation. It could be speculated, that the absence of other closed-shell HOMs from HOM-RO$_2$ + NO reactions, other than organic nitrates, is probably caused by preferred alkoxy decomposition into C$_2$- and C$_3$-fragments, which are hard to detect with the used analytical technique.

Experiments in dependence of relative humidity (r.h.) showed no clear trend of product formation with rising r.h. (Supplementary Fig. 10), which indicates the absence of a significant effect of water vapor on the HOM formation. H/D-exchange experiments in the presence of heavy water confirmed the total number of OOH and OH groups in the products as expected from the proposed reaction scheme, i.e., two groups with exchangeable H atoms were identified in the case of C$_4$H$_6$O$_6$ and C$_5$H$_9$O$_5$NO$_2$ and three for C$_5$H$_9$O$_8$, C$_5$H$_9$O$_9$, C$_5$H$_9$O$_7$NO$_2$ and C$_5$H$_9$O$_8$NO$_2$ (Supplementary Fig. 11).

Concentrations of major HOM products depending on added NO are depicted in Fig. 7 from experiments applying TME ozonolysis for OH generation and results obtained by means of IPN photolysis in Supplementary Fig. 12. C$_5$H$_9$O$_8$, the organic nitrate C$_5$H$_9$O$_7$NO$_2$ and C$_4$H$_6$O$_6$ dominate the HOM formation for NO concentrations ≥ 10$^{10}$ molecules cm$^{-3}$ qualifying δ-RO$_2$ + NO as an important channel for HOM generation under conditions with elevated NO concentrations. The simultaneous increase of C$_4$H$_6$O$_6$ along with C$_5$H$_9$O$_7$NO$_2$ supports that C$_4$H$_6$O$_6$ is formed from the C$_5$H$_9$O$_8$ + NO reaction (Fig. 7 and Supplementary Fig. 12).

The RO$_2$ radicals C$_5$H$_9$O$_8$ **27** and C$_5$H$_9$O$_9$ **19** represent the end-point of the autoxidation chain of the respective reaction channels after three autoxidation steps each. Still higher oxidized compounds were not detectable, even not in traces, and their possible formation pathways would be difficult to explain mechanistically. The amount of produced "final" HOM-RO$_2$ radicals C$_5$H$_9$O$_8$ and C$_5$H$_9$O$_9$ were determined from their measured concentrations considering the reaction with NO as the only important loss process, see Eq. (13) in Methods. These data can be taken as a conservative estimate for the formation of HOM-RO$_2$ radicals and maximum closed-shell HOM products independent of further reactions of the HOM-RO$_2$ radicals, see results from experiments using TME ozonolysis in Supplementary Fig. 13. The relatively small importance of other HOM products with less oxygen content, i.e., C$_5$H$_9$O$_6$ and C$_5$H$_9$O$_7$ radicals and their closed-shell products, justifies this approach (Fig. 6).

Resulting molar HOM-RO$_2$ production yields from experiments with both OH radical sources were in reasonable agreement (Fig. 8). It is to be noted, that the amount of non-converted HO-C$_5$H$_8$O$_2$ radicals, determined from modeling, was considered in the normalization with respect to reacted isoprene. (Non-converted) HO-C$_5$H$_8$O$_2$ radicals represented the main product portion from OH + isoprene for low NO conditions, which would clearly distort the product yields applied for atmospheric conditions due to the restricted residence time of 32 s in the experiments not allowing adequate HO-C$_5$H$_8$O$_2$ conversion. For C$_5$H$_9$O$_9$, a production yield of about $0.3^{+0.3}_{-0.15}$ % was obtained for NO concentrations ≤ 10$^9$ molecules cm$^{-3}$ decreasing to $0.08^{+0.08}_{-0.04}$ % for ~10$^{10}$

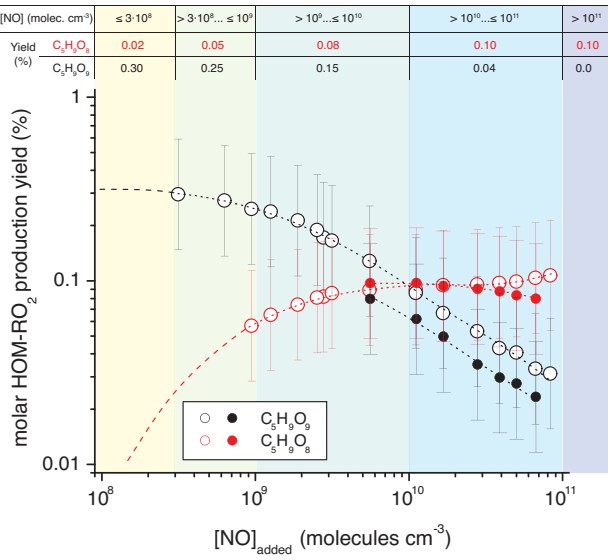

| [NO] (molec. cm⁻³) | ≤ 3·10⁸ | > 3·10⁸... ≤ 10⁹ | > 10⁹...≤ 10¹⁰ | > 10¹⁰...≤ 10¹¹ | > 10¹¹ |
|---|---|---|---|---|---|
| Yield (%) $C_5H_9O_8$ | 0.02 | 0.05 | 0.08 | 0.10 | 0.10 |
| $C_5H_9O_9$ | 0.30 | 0.25 | 0.15 | 0.04 | 0.0 |

**Fig. 8 | Molar production yields of the HOM-RO₂ radicals $C_5H_9O_8$ and $C_5H_9O_9$ depending on NO concentration.** (HOM highly oxygenated molecule) Open symbols show the results from TME (tetramethylethylene) ozonolysis for OH generation and closed symbols from IPN (isopropyl nitrite) photolysis. HOM-RO₂ measurements were taken from the experiments given in Fig. 7 and Supplementary Fig. 12 as well as in Supplementary Fig. 4, but here using nitrate ionization for product detection. The amount of produced HOM-RO₂ radicals, determined from Eq. (13), was normalized by the reacted isoprene taken from modeling (Methods). A correction with respect to non-converted HO-C₅H₈O₂ radicals has been done. Calculated background HO₂ concentrations were in the range (1–3) × 10⁸ (TME ozonolysis) and (7–12) × 10⁸ molecules cm⁻³ (IPN photolysis). Error bars represent the uncertainty of a factor of ∼2 in the measured concentrations. The dashed lines show assumed extrapolations. Stated average yields for given NO ranges were used in the global modeling.

molecules cm⁻³ mainly due to repression of the 1,6 H-shift isomerization with rising NO. The $C_5H_9O_8$ production yield increased with rising NO up to about $0.1^{+0.1}_{-0.05}$ % for [NO] ∼10¹⁰ molecules cm⁻³ due to rising importance of the δ-RO₂ + NO channel and remained almost unchanged for higher NO. This behavior indicates rapid RO₂ isomerization in this system being competitive with the RO₂ + NO rates, e.g., k(**22**→**26**) vs. k(**22** + NO) × [NO] and k(**26**→**27**) vs. k(**26** + NO) × [NO] (Fig. 3). This analysis shows that HOM-RO₂ radicals in total were produced with molar yields of about 0.13–0.35 % in the considered NO range of ∼10⁹ up to 8.3 × 10¹⁰ molecules cm⁻³, only slightly repressed by higher NO levels.

The data analysis allowed an estimate of the molar organic nitrate yield from RO₂ + NO reactions supposing identical detection efficiency for the respective RO₂ radical and RONO₂ (Eq. (15) in Methods); molar RONO₂ yield: 22 ± 4 % for $C_5H_9O_8$ and 19 ± 3 % for $C_5H_9O_9$ considering the statistical uncertainty only (Supplementary Fig. 14). A molar $C_4H_6O_6$ yield of 17 ± 4 % from the $C_5H_9O_8$ + NO reaction followed based on the measured product ratio $[C_4H_6O_6] / [C_5H_9O_7NO_2] = 0.78 ± 0.09$ (Supplementary Fig. 15) assuming identical detection efficiency again. Thus, in the atmosphere, total closed-shell HOM yields of 39 ± 8 % and 19 ± 3 % can be expected from the reaction of $C_5H_9O_8$ and $C_5H_9O_9$ with NO, respectively. In the case of the HOM-RO₂ + HO₂ reaction the situation is not yet clear. While for simple RO₂ radicals, such as for HO-C₅H₈O₂, molar ROOH yields of close to unity have been reported[4,9], clearly smaller ROOH yields are proposed for higher oxidized RO₂ radicals bearing especially carbonyl and OOH groups in the vicinity of the peroxy group[4,9]. Experimental data for this, however, are barely available. Thus, closed-shell HOM yields from the reaction of $C_5H_9O_8$ and $C_5H_9O_9$ with HO₂ radicals would be speculative at the moment and are not stated here. Attempts to determine them experimentally failed up to now.

## Atmospheric impact

The chemistry climate model ECHAM-HAMMOZ was applied to simulate the global impact especially of the HOM formation. The simulations were performed for the year 2010 with a three months spin-up time, a spatial horizontal resolution of 1.875° × 1.875° and 47 vertical layers (up to 0.01 hPa height). Isoprene emission was calculated interactively by using the biogenic emission module MEGAN[37,38]. The initial, fixed RO₂ radical distribution in the chemistry model MOZ[39,40] has been adjusted to fit the observed MVK and MACR yields in experiments with atmospherically relevant NO levels of (5–20) × 10⁹ molecules cm⁻³ given in the literature[12,41]. Accordingly, the branching of lumped HO-C₅H₈O₂ radicals was set (model species names) LISOPACO₂ / ISOPBO₂ / ISOPDO₂ = 17 % / 50 % / 33 %, where LISOPACO₂ stands for all δ-RO₂ radicals, ISOPBO₂ for the β-RO₂ radical **5** and ISOPDO₂ for the β-RO₂ radical **5´**.

At first, the contributions of the reactions of initially formed HO-C₅H₈O₂ radicals either with (i) HO₂, (ii) NO, (iii) CH₃O₂ and CH₃C(O)O₂ (further denoted as RO₂) or (iv) via 1,6 H-shift isomerization were analyzed. The simulation revealed an annual mean channel ratio (i) / (ii) / (iii) / (iv) of 47 % / 34 % / 4 % / 14 % in line with previous results[16,42]. Spatial distribution of the annual mean HO-C₅H₈O₂ reaction rates (vertical sum for each grid cell) is provided in Supplementary Fig. 16.

Secondly, the annual HOM-RO₂ radical production for $C_5H_9O_8$ and $C_5H_9O_9$ was calculated offline using their experimentally obtained NO-dependent molar formation yields including the extrapolated values (Fig. 8). Each grid cell for each time step was treated separately in the summation. According to this, 0.5 × 10⁶ metric tons of $C_5H_9O_8$ radicals and 3.8 × 10⁶ metric tons of $C_5H_9O_9$ radicals are produced annually. The simulation clearly showed that $C_5H_9O_9$ radical formation from the 1,6 H-shift isomerization channel mainly relates to the low-NO regime in the tropics with the highest isoprene emission, whereas the δ-RO₂ + NO channel related $C_5H_9O_8$ radical formation is more spatially distributed (Fig. 9).

The atmospheric lifetime of RO₂ radicals, including both HOM-RO₂ radicals, is in the order of a few tens of seconds or less due to their reactions with NO or HO₂ radicals[4]. Thus, as the next step, the further reactions of $C_5H_9O_8$ and $C_5H_9O_9$ radicals with NO and HO₂ were considered in the global modeling. According to this, 61 % of $C_5H_9O_8$ radicals are reacting with NO and 39 % with HO₂ and in the case of $C_5H_9O_9$ radicals 46 % with NO and 54 % with HO₂. This shows that a substantial fraction of HOM-RO₂ radicals are reacting with HO₂ in the atmosphere. However, the closed-shell HOM yield from HOM-RO₂ + HO₂ is highly speculative at the moment due to the lack of experimental information as mentioned before. Thus, it can be only stated, that an annual global closed-shell HOM production in the order of (1–2) × 10⁶ metric tons can be expected based on the corresponding total HOM-RO₂ radical production of about 4 × 10⁶ metric tons.

For comparison, as the result of a former study, carried out in the LFT under low-NO conditions for elevated isoprene conversion[22], a molar HOM yield of $0.03^{+0.03}_{-0.015}$ % (RO₂ radicals and closed-shell products in total) from OH + isoprene was reported being one order of magnitude smaller than the finding of the present study. Main reasons for this discrepancy are (i) that the fraction of non-converted HO-C₅H₈O₂ radicals was not considered in the reported yield and (ii) significant RO₂ + RO₂ reactions lowering the HOM-RO₂ formation yield. On the other hand, a HOM yield of up to 11 % as proposed from theoretical calculations for low-NO conditions[17] appears to be definitely too high compared to the experimentally based findings.

The isoprene-derived HOM-RO₂ formation was checked against the corresponding processes starting from the ozonolysis and OH radical reaction of α-pinene, which amounts to about one-third of total monoterpene emission[2]. The oxidation of α-pinene is currently considered as a very important HOM-generating process in the atmosphere[24,43]. Needed modeling calculations were carried out for the year 2010 in an analogous way as described before for OH +

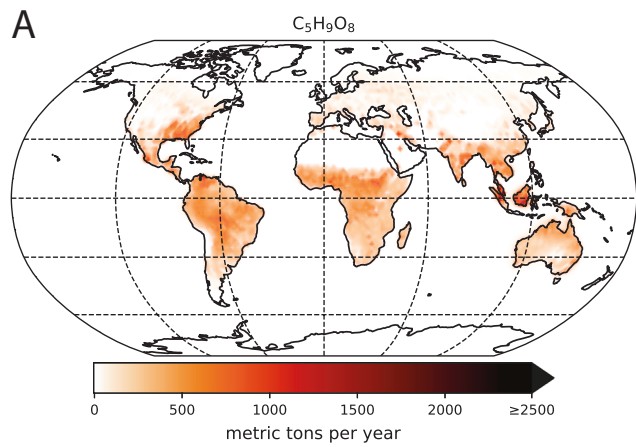

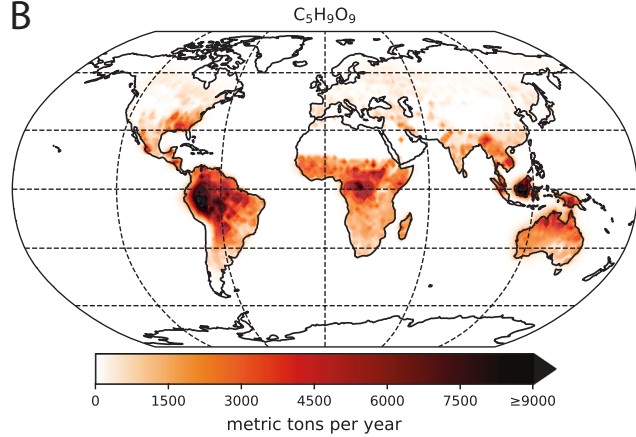

**Fig. 9 | Spatial distribution of HOM-RO₂ radical production from OH + isoprene.** (HOM highly oxygenated molecule) Annual total production in each grid cell for **A** $C_5H_9O_8$ from the δ-RO₂ + NO channel and **B** $C_5H_9O_9$ from the 1,6 H-shift isomerization channel. The data were plotted with Python using the packages cartopy (projection: Robinson)[64] and matplotlib (https://matplotlib.org/)[65,66].

isoprene using ECHAM-HAMMOZ including the emission module MEGAN[37,38]. $C_{10}H_{15}O_8$ and $C_{10}H_{15}O_{10}$ (in the following summarized as $C_{10}H_{15}O_{8,10}$) were considered as the important HOM-RO₂ radicals from α-pinene ozonolysis[22,34,43–46] and $C_{10}H_{17}O_7$ from the OH radical reaction[33,47–50]. NO-dependent molar production yields of $C_{10}H_{15}O_{8,10}$ and $C_{10}H_{17}O_7$ were determined as explained for the isoprene-related HOM-RO₂ radicals using experimental data as recently reported (Supplementary Fig. 17)[33,34]. For low-NO conditions, molar production yields of $6.9^{+6.9}_{-3.5}$ % for $C_{10}H_{15}O_{8,10}$ and $7.6^{+7.6}_{-3.8}$ % for $C_{10}H_{17}O_7$ were obtained being a factor of about 20 higher compared with the results from isoprene oxidation. The α-pinene-derived molar yields are at the upper end or significantly higher compared to the reported HOM data (mainly for closed-shell HOMs) measured by means of nitrate ionization[24]. It is to be noted here, that the nitrate ionization scheme appears to be less sensitive especially for $C_{10}H_{17}O_7$ and subsequent products[33,47].

The modeling results show an annual production of $3.6 \times 10^6$ metric tons of α-pinene-derived HOM-RO₂ radicals, $2.0 \times 10^6$ metric tons of $C_{10}H_{15}O_{8,10}$ and $1.6 \times 10^6$ metric tons of $C_{10}H_{17}O_7$, very similar to the total HOM-RO₂ production of $4.3 \times 10^6$ metric tons from OH + isoprene (Table 1). Spatial distribution of the annual production of $C_{10}H_{15}O_{8,10}$ and $C_{10}H_{17}O_7$ is shown in Supplementary Fig. 18 and the fraction of their subsequent reactions either with HO₂ or NO in Table 1.

The similar HOM-RO₂ radical production starting from isoprene and α-pinene on global scale is caused by the huge isoprene emission, about 20-fold that of α-pinene[2], which compensates the much lower HOM-RO₂ formation yields for isoprene with about only one twentieth compared to the one of α-pinene, cf., Fig. 8 and Supplementary Fig. S17. It should be mentioned at this point, that HOM formation in the isoprene system can also arise from secondary chemistry of isoprene products, i.e., from the reaction of OH radicals with hydroxy hydroperoxides $C_5H_{10}O_3$ forming especially dihydroxy dihydroperoxides $C_5H_{12}O_6$[4,29,51]. The formation pathway leading to $C_5H_{12}O_6$ and its importance in the atmosphere, however, are currently controversially discussed[52,53].

In conclusion, we experimentally showed the formation of a series of oxidized C₄- and C₅-products with at least two functional groups from the OH + isoprene reaction for conditions of the equilibrated HO-$C_5H_8O_2$ radical distribution being relevant in the atmosphere. NO levels covered the range (< 2–830) × 10⁸ molecules cm⁻³ characterizing the situation in remote to urban areas. For background NO conditions < 2 × 10⁸ molecules cm⁻³, a $C_5H_8O_2$ product was detected, whose structure and possible formation routes are unclear at the moment. All other observed products could be assigned either to the 1,6 H-shift channel (Fig. 2) or the δ-RO₂ + NO channel (Fig. 3). A couple of them are reported for the first time. The structural similarity of allyl radicals **10** and **11** (Fig. 1) from the respective channels causes a similar chemical behavior connected with comparable product formation.

HOM formation was observed from both reaction channels forming $C_5H_9O_9$ **19** and $C_5H_9O_8$ **27** as the "final" HOM-RO₂ radicals, respectively, after three autoxidation steps. Although there exists lack of knowledge regarding the product yield from the HOM-RO₂ + HO₂ reaction, annual closed-shell HOM production in the order of (1–2) × 10⁶ metric tons globally can be expected from the OH + isoprene reaction representing an additional HOM source to the important one from the oxidation of α-pinene[24].

All in all, the results of the present study provide a more complete mechanistic insight into isoprene's atmospheric degradation process and the resulting product formation. The detected HOM formation from both reaction channels, comparable with that of the α-pinene oxidation, allows a better understanding of the formation of SOA precursors in the atmosphere.

## Methods
### Experimental setup
The experiments were carried out in the laminar flow tube (LFT)[29,30] operating at 1 bar of air at a temperature of 295 ± 0.5 K and a relative humidity (r.h.) of < 0.1 % by default. R.h. was increased up to 28.6 % in humidity-dependent measurements. The reaction time in the LFT was 32 s experimentally obtained by a "chemical clock". Ozonolysis of tetramethylethylene (TME)[54] as well as photolysis of isopropyl nitrite (IPN)[55] or H₂O₂ served as the OH radical source.

The laminar flow tube (LFT) (i.d. 8 cm; total length 425 cm) consists of a first section (56 cm) containing the gas inlet system, a second

**Table 1 | Global HOM-RO₂ production and their fraction reacting with HO₂ and NO**

| HOM-RO₂ radicals | annual production (10⁶ metric tons) | reacting with HO₂ (%) | reacting with NO (%) |
|---|---|---|---|
| $C_5H_9O_8$ | 0.5 | 39 | 61 |
| $C_5H_9O_9$ | 3.8 | 54 | 46 |
| $C_{10}H_{15}O_{8,10}$ | 2.0 | 44 | 56 |
| $C_{10}H_{17}O_7$ | 1.6 | 23 | 77 |

*HOM* highly oxygenated molecule.

middle section (344 cm) representing the reaction zone surrounded by 8 UV lamps (Hg-lamps made of PN235 quartz-glass with a cut-off wavelength of 210 nm) and 24 Philips 36 W Blacklight Blue lamps emitting in the range 350–400 nm. An end section (25 cm) incorporating the sampling devices. Applying TME ozonolysis, ozone was injected through a nozzle system into the gas mixture, including TME and isoprene, just before entering the middle section. In the case of photolysis experiments, the respective OH precursor was added to the main gas stream containing isoprene. The total flow was set at 30 L min$^{-1}$ (STP).

The concentrations of the organic reactants were monitored by means of a proton transfer reaction mass spectrometer at the outflow of the flow systems (Ionicon, PTR-MS 500)[56]. The relative humidity of the reaction gas was continuously controlled by a humidity sensor (Hygrosense HYTE) and ozone was monitored by a gas monitor (Thermo Scientific, iQ 49).

All gas flows were set by means of calibrated gas flow controllers (MKS 1259/1179). The organic reactants and gases had the following purity: isoprene (>99 %, Aldrich), isoprene-1-$^{13}$C (≥ 97 %, isotopic purity: ≥ 99 % $^{13}$C, Aldrich), TME (≥ 99 %, Aldrich) and NO (498 ± 10 ppmV NO (99.5 %) in N$_2$ (99.999 %), Air Liquide). Air was taken from a commercial PSA (Pressure Swing Adsorption) unit with further purification by a series of absorber units filled with hopcalite (CuMnOx catalyst), activated charcoal and different 4 Å and 10 Å molecular sieves.

## Product analysis by mass spectrometry

Detection of isoprene´s oxidation products was performed by means of a CI-APi-TOF (chemical ionization – atmospheric pressure interface – time-of-flight) mass spectrometer (Tofwerk AG, Airmodus) with a resolving power > 3000 Th/Th sampling from the center flow of both flow system with a sampling rate of 10 L min$^{-1}$ (STP). The ion-molecule reaction (IMR) took place at atmospheric pressure using a Boulder-type inlet[57]. As the reagent ions in the product measurements served ethylaminium (C$_2$H$_5$NH$_3$$^+$), iodide (I$^-$) and nitrate (NO$_3$$^-$). Ammonium (NH$_4$$^+$) was used in additional calibration experiments.

In the case of ionization by ammonium, a flow of 2 ml nitrogen over a ∼2 percent NH$_3$ / H$_2$O solution was added to a 35 L min$^{-1}$ (STP) sheath flow of high-purity nitrogen forming NH$_4$$^+$ after ionization with a $^{241}$Am source. The ions from the sheath flow were guided into the sample flow by an electric field without mixing of both gas streams. In the case of ionization by ethylaminium, ethylamine premixed in a flask was added to the sheath flow resulting in an amine concentration of 2.4 × 10$^{11}$ molecules cm$^{-3}$. Formed reagent ions were C$_2$H$_5$NH$_3$$^+$ and to a lesser amount (C$_2$H$_5$NH$_2$)C$_2$H$_5$NH$_3$$^+$. For ionization by iodide, the 35 L min$^{-1}$ (STP) sheath flow contained 5 × 10$^{11}$ molecules cm$^{-3}$ of tert-butyl iodide and the resulting reagent ions were I$^-$ and traces of I(H$_2$O)$^-$. In the case of ionization by nitrate, an HNO$_3$ containing vial was connected to the sheath flow without overflowing the HNO$_3$ sample. HNO$_3$ diffusion from the vial was found to be sufficient to form the reagent ions (HNO$_3$)$_x$NO$_3$$^-$, $x$ = 0, 1, 2, after ionization.

RO$_2$ radicals and closed-shell products were detected as a cluster with the respective reagent ions. Stated normalized product signals were determined according to Eqs. (1)–(3) using either ethylaminium (C$_2$H$_5$NH$_3$$^+$), iodide (I$^-$) or nitrate (NO$_3$$^-$), respectively.

$$Normalized\ signal = \frac{[(product)C_2H_5NH_3^+]}{[C_2H_5NH_3^+] + [(C_2H_5NH_2)C_2H_5NH_3^+]} \quad (1)$$

$$Normalized\ signal = \frac{[(product)I^-]}{[I^-]} \quad (2)$$

$$Normalized\ signal = \frac{[(product)NO_3^-]}{[NO_3^-] + [(HNO_3)NO_3^-] + [(HNO_3)_2NO_3^-]} \quad (3)$$

In the case of ammonium ionization, the determination of the normalized signal intensity was impossible because the ammonium signal at nominal 18 Th could not be measured with the needed reliability. The quantities in Eqs. (1)–(3), i.e., [(product)C$_2$H$_5$NH$_3$$^+$] etc., are the measured signal intensities. Duty cycle correction is applied in order to compensate for the mass-dependent transmission of the mass spectrometer[14]. Stated results were obtained from 10 min data accumulation. Statistical uncertainty of signal intensities was less than 20 %, in most cases less than 10 %.

The product concentrations [product] can be calculated from the normalized product signals, Eqs. (1)–(3), multiplied by the estimated calibration factor f, e.g., for ionization by ethylaminium:

$$[product] = f(C_2H_5NH_3^+, product) \frac{[(product)C_2H_5NH_3^+]}{[C_2H_5NH_3^+] + [(C_2H_5NH_2)C_2H_5NH_3^+]}$$
$$(4)$$

Analogous equations result for iodide and nitrate starting from Eqs. (2) and (3). Calculated lower limit calibration factors are in the range $f_{calc}$ = (1.3–2.8) × 10$^9$ molecules cm$^{-3}$ assuming (i) collision limit in the ion-molecule reaction and (ii) no ion loss in the instrument[14]. Here, $f_{calc}$ = 2 × 10$^9$ molecules cm$^{-3}$ with an uncertainty of a factor of 2 was taken in line with our recent absolute H$_2$SO$_4$ calibration using nitrate ionization where requirements (i) and (ii) seem to be fulfilled[58].

Comparison of lower limit concentrations for a series of reagent ions with raising sensitivity, supported by theoretical calculations on the cluster stability of (product)reagent-ions, allowed to conclude that products of the OH + isoprene reaction bearing a hydroperoxy group and at least one hydroxy or an additional hydroperoxy or carbonyl group, i.e., C$_5$H$_8$O$_3$, C$_5$H$_8$O$_4$, C$_4$H$_8$O$_5$, and C$_5$H$_9$O$_5$, are detectable with close to maximum sensitivity using either hydrazinium or methylaminium, see Figs. 4c, S4, S5 and 4b, respectively, in ref. 14. Thus, the measured lower limit concentrations obtained with these ionization schemes should be close to the "real" concentrations applying the lower limit calibration factor $f_{calc}$[14,15]. Measurements using ammonium supported the conclusion. Based on this, calibration factors for product measurements by ethylaminium and iodide ionization were estimated (Supplementary Table 2)[14,15]. In the case of C$_5$H$_8$O$_2$, close to maximum detection sensitivity was probably not fulfilled even in the case of ammonium ionization (Supplementary Fig. 19). According to this, only lower limit concentrations can be stated. The calibration factor of C$_4$H$_8$O$_4$ was assumed to be the same as for C$_4$H$_8$O$_5$ and for C$_4$H$_8$O$_2$ the same as for C$_5$H$_8$O$_2$. For HOM measurements by means of nitrate ionization $f$ = 2 × 10$^9$ molecules cm$^{-3}$ was used throughout.

## Determination of reacted isoprene

Direct measurement of the isoprene conversion was experimentally impossible because of the restricted isoprene conversion of less than 8 × 10$^9$ molecules cm$^{-3}$, which accounts for less than 1 % (mostly less than 0.1 %) of the initial isoprene concentration. Therefore, the amount of reacted isoprene was calculated using a complex reaction scheme including the initial RO$_2$ radical processes[4,7], see Supplementary Table 1. This reaction scheme was also used in order to determine the reaction flux over the 1,6 H-shift and the δ-RO$_2$ + NO channel and to calculate the HO$_2$ radical level in the experiments.

Moreover, in the case of OH radical generation from IPN photolysis, SO$_2$ was added in small quantities, not disturbing the product formation from OH + isoprene. From the measured SO$_3$ formation, OH + SO$_2$ ( + O$_2$) → SO$_3$ + HO$_2$, k(295 K) = 8.9 × 10$^{-13}$ cm$^3$ molecules$^{-1}$ s$^{-1}$[59], the average OH radical concentration in the LFT was determined, which allowed to calculate the converted isoprene for a given isoprene concentration[15].

## Wall loss in the LFT

The rate law for any formed product Prod is given by

$$\frac{d[Prod]}{dt} = P_{Prod} - k_{loss}[Prod] \qquad (5)$$

applying the time-independent production term of Prod, $P_{Prod}$. This assumption is justified because of constant OH production during the whole reaction time and practically constant isoprene concentrations due to reactant conversion smaller than 1 % in each case. Integration of Eq. (5) with $[Prod]_{t=0} = 0$ yields:

$$[Prod]_t = \frac{P_{Prod}}{k_{loss}} (1 - \exp(-k_{loss}t)) \qquad (6)$$

$[Prod]_t = P_{Prod} t$ follows for the wall-loss free Prod concentration. Consequently, the relative Prod loss in the tube is given by:

$$Prod\ loss = 1 - \frac{1}{k_{loss}t} (1 - \exp(-k_{loss}t)) \qquad (7)$$

The value of $k_{loss}$ can be described by the diffusion-controlled wall-loss term $\frac{3.65 \times D}{r^2}$ using an average diffusion coefficient for the isoprene products $D = 0.08\ cm^2\ s^{-1}$[60] leading to $k_{loss} = 0.018\ s^{-1}$. Based on that and for the reaction time of 32 s in the LFT, a product loss of 24 % was calculated using Eq. (7). The measured Prod concentrations were corrected accordingly.

## Kinetic data analysis

Production of "final" HOM-RO₂ radicals $C_5H_9O_8$ and $C_5H_9O_9$:

The rate law for the "final" HOM-RO₂ radicals is given by:

$$\frac{d[HOM-RO_2]}{dt} = P_{HOM-RO_2} - k_{RO_2+NO} \times [HOM-RO_2] \times [NO] \qquad (8)$$

"$P_{HOM-RO_2}$" stands for the production term of $HOM-RO_2$ radicals, not further specified. The production term can be considered as time-independent because of practically constant reactant concentrations with time for the chosen conditions. This leads to an (almost) linear signal rise of RO₂ radicals with time as experimentally shown in a similar reaction system[33]. Thus, "$[HOM-RO_2]$" can be explained via the relationship in Eq. (9).

$$[HOM-RO_2] = f(t) = a \times t, a = constant \qquad (9)$$

Insertion of Eq. (9) into Eq. (8) yields:

$$\frac{d[HOM-RO_2]}{dt} = P_{HOM-RO_2} - k_{RO_2+NO} \times [NO] \times a \times t \qquad (10)$$

Integration of Eq. (10) with $[HOM-RO_2] = 0$ at $t = 0$ and $[NO]_t = [NO]_0$ leads to:

$$[HOM-RO_2]_t = \int_0^t P_{HOM-RO_2} dt - 0.5 k_{RO_2+NO} \times [NO]_0 \times a \times t^2 \qquad (11)$$

Make use of Eq. (9) in Eq. (11), it follows:

$$[HOM-RO_2]_t = \int_0^t P_{HOM-RO_2} dt - 0.5 k_{RO_2+NO} \\ \times [NO]_0 \times [HOM-RO_2]_t \times t \qquad (12)$$

Thus, the integral HOM-RO₂ formation as a function of NO for a given time t can be given as:

$$\int_0^t P_{HOM-RO_2} dt = [HOM-RO_2]_t \times \left(1 + 0.5 k_{RO_2+NO} \times [NO]_0 \times t\right) \qquad (13)$$

Although $C_5H_9O_8$ **27** and $C_5H_9O_9$ **19** are expected to be produced with an acyl RO₂ structure, very rapid 1,6 and 1,7 H-shifts from the OOH groups[36,61] immediately form the corresponding non-acyl RO₂ radicals making the universal rate coefficient $k(RO_2 + NO) = 8.8 \times 10^{-12}\ cm^3$ molecule$^{-1}$ s$^{-1}$, T = 295 K, applicable in Eq. (13)[4,9].

Formation yield of organic nitrates $C_5H_9O_7NO_2$ and $C_5H_9O_8NO_2$: The rate law of RONO₂ formation reads as follows:

$$\frac{d[RONO_2]}{dt} = y_{RONO_2} \times k_{RO_2+NO} \times [RO_2] \times [NO] \qquad (14)$$

The term $y_{RONO_2}$ stands for the formation yield of the organic nitrates. Integration of differential Eq. (14) with $[RONO_2] = 0$ at $t = 0$ and $[NO]_t = [NO]_0$ applying again the relationship in Eq. (9) leads to:

$$[RONO_2]_t = 0.5 \times y_{RONO_2} \times k_{RO_2+NO} \times [RO_2]_t \times [NO]_0 \times t \qquad (15)$$

## Data availability

The experimental data generated in this study are provided in the paper and in the Supplementary Information. The model data that support the findings of this study have been deposited in the public research data archive ZENODO (10.5281/zenodo.14389292)[62].

## Code availability

The ECHAM-HAMMOZ model (https://redmine.hammoz.ethz.ch/projects/hammoz)[63] is developed by a consortium composed of ETH Zurich, Max Planck Institut für Meteorologie, Forschungszentrum Jülich, University of Oxford, the Finnish Meteorological Institute and the Leibniz Institute for Tropospheric Research, and managed by the Leibniz Institute for Tropospheric Research (TROPOS). The ECHAM-HAMMOZ model source code and all required input data are freely available after signature of a license agreement.

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

## Acknowledgements
The authors thanks R. Gräfe and A. Rohmer for technical assistance and the tofTools team for providing the data analysis tools. T.B. thanks P. O. Wennberg and J. D. Crounse, Caltech, for helpful discussions. This work used resources of the Deutsches Klimarechenzentrum (DKRZ) granted by its Scientific Steering Committee (WLA) under project ID bb1128, E.H.H.

## Author contributions
T.B. designed and carried out the experiments and did the data analysis. E.H.H., A.T. and H.H. conducted the global modeling work. T.B., E.H.H. and A.T. wrote the draft and all authors contributed to the final version of the manuscript.

## Funding

## Competing interests
The authors declare that they have no competing interests.
