## [Transparent Peer Review file · Nature Communications]

Highly oxidized products from the atmospheric reaction of hydroxyl radicals with isoprene

Corresponding Author: Dr Torsten Berndt

Version 0:

Reviewer comments:

Reviewer #1

(Remarks to the Author)

The study by Berndt et al. focuses on the investigation of the oxidation of isoprene by OH radicals at atmospheric like conditions with experiments carried on in a laminar flow tube and by measuring oxidation products with a chemical ionization – atmospheric pressure interface – time-of-flight (CI-API-TOF) instrument.

The focus of the study is the quantification of peroxy radicals which are precursors of highly oxidized molecules (HOMs) which contribute to formation of particles. Based on this study, these radicals are formed with a molar yield of about 0.35% at ambient-like concentrations of NO.

As per usual from previous studies from this group, the manuscript is well written and structured and mostly easy to follow. The chemical paths suggested for the observed radicals and closed-shell species are justified by literature and/or represent a reasonable assumption based on similar molecules. I believe in the capability of this group in detecting and separating such reactive radicals so I do not have any reason to doubt the findings.

What is not so clear to me is what is the high impact of the study to justify publication on Nature Communications. As pointed out by the authors, the yields found in this study are one order of magnitude higher than what previously reported for studies performed at high concentrations which might not be representative of ambient conditions. Still, the yield remains relatively small. I understand that isoprene emissions are very large so such a small yield can still have a large impact but from the current version of the manuscript it is not clear what the impact is. The authors include a model run and give numbers for the production rate of the radicals they identified but what I am missing is a comparison with other known sources of HOMs. It is mentioned in the paper that these new paths and radicals would contribute in addition to the important source represented by monoterpenes, but no comparison is given so it is hard to judge how much of a contribution this is. Are the authors suggesting that they should be included in global models to bring measurements and modelled results into agreement? It could also be that just the fact that they are able to detect and identify these radicals is per se an excellent achievement so I would suggest stressing this more in the manuscript.

My major point is therefore to clearly show how much this additional source of HOMs contributes to the current known sources. My recommendation would be to run the global model with the current known sources and add the new yield for isoprene found in this study and see what number (percentage ideally) comes out.

Minor comment:

Title is a bit weird as with higher I would expect a comparison to show up...higher than previously expected ... for example. As it is now, I am left wondering higher than what?

Reviewer #2

(Remarks to the Author)

This study by Berndt and colleagues present study on the yield of HOM molecules produced from OH initiated oxidation of isoprene under atmospherically relevant conditions of concentrations for RO₂ and NO. This study pinpoints the structure of molecules that would be dominant for HOMs and the mechanisms for production of such molecules particularly C₅H₉O₈ and C₅H₉O₉. Compared to previous study on a similar topic by Wei Nei et al Nat. Comm. 14, 3347, 2023, the mechanistic study presented here is significantly more focused and robust. Considering that isoprene chemistry plays such a large role in the atmosphere, I would support publication of this study in Nature Communications after the authors have responded to the following comments.

Major comments

-C5H8O2 yield is comparable to the analogue HPALD C5H8O3 even under low NO conditions which seems a bit high. Can the author comment on how fast the alkoxy 1,5 H shift would have to be to account for this? Or are there any other RO2 in the system such as CH3O2 which can form alkoxy in their apparatus?

-Medeiros and co-authors (JACS Au 2022, 2, 809-818) have recently suggested that there is a significant OH production from dihydrocarbonyl pathways which would have been verifiable from this study. However, the yield of the co-C4H8O5 product in that pathway is measured to be low here and seems to be in contradiction. The authors should comment on this.

-Is there any reason why NO cannot be quantified below 2×10^8 molecule cm^{-3} concentration?

Minor comments

Bring Figure S1 to the main text

Line 34, change 'emphasis' to 'emphasize'

Line 42, change 'leads to a variety' to 'leads to production of a variety'

Line 68, change 'Predominate' to 'The main products'

Line 84, delete 'also' change 'as well as similar' to 'and'

Line 112, reword to 'allyl radical 11 can be produced via the alkoxy channels of the reactions between 4 or 6 with other RO2 radicals'

Line 125, delete 'at the minimum'

Line 131, change 'being' to 'which are'

Line 193, change 'substance flow' to 'reaction flux'

Line 253-54, this sentence needs better wording

Line 258, change 'definitionally' to 'by definition'

Line 359, check '17+/-4' to '19+/-3'

Line 404, delete 'finally important for SOA generation'

Line 479, reword

Line 568, its not obvious where 0.5 comes from

Reviewer #3

(Remarks to the Author)

Berndt et al. present a very nice detailed mechanistic study on the formation of a complex range of OH + isoprene products (some for the first time) that can lead to highly oxygenated molecular precursors to secondary organic aerosol, which have been assigned to different RO2 production channels, one which is prevalent under pristine atmospheric conditions and one which is potentially active under moderate NO conditions (at least as shown under the conditions of the experiments). The paper is well written and the experimental approach is reasonably sound given the complexities of the system being studied and the low concentrations of the species under investigation. There is a large amount of uncertainties in these experiments, as highlighted by the authors, leading to some speculation in the impacts of the results. It is the potential importance of these results on global atmospheric chemistry which is not what I am really getting from the way they are presented here. How does the potential to impact biogenic SOA formation compare to what the regional and global models already predict coming from other, more high yielding isoprene product precursor species such as, for example, isoprene epoxides (IEPOX) (low NO pathways) or e.g. 2-methylglyceric acid (high NO pathways)? How do these new potential SOA formation pathways compare to those from monoterpenes (these routes briefly mentioned in the paper)? For this to be a Nature Communications paper, I would like to see more discussions on the potential regional and global impacts of these results. Having said that, as the paper is written, it would make a solid publication in a more atmospheric chemistry focused journal as the results will be very interesting to atmospheric chemical mechanism developers and modellers.

Some more specific point are given below:

The mechanistic background to this work is mainly focused around that of the Wennberg et al., 2018 review, and some subsequent studies after this. However, how do the pathways presented here compare to those given in other detailed isoprene mechanisms used extensively in atmospheric models, such as the MCMv3.3.1 isoprene scheme (Jenkin et al., 2015 (www.atmos-chem-phys.net/15/11433/2015/)) and the new Julich isoprene mechanisms (e.g. Tsiligiannis et al., 2022; <https://doi.org/10.1029/2021gl097366>), whose initial OH-isoprene adducts with O2 chemistry seems more complex than that presented here?

MCMv3.3.1 website is referenced (24). It would be better to reference the paper that is focused on the MCMv3.3.1 isoprene mechanism update instead, i.e. Jenkin et al., 2015 (www.atmos-chem-phys.net/15/11433/2015/).

Throughout the text the emissions are given in metric tonnes per year. Please convert to more conventional units in order to compare to other global modelling studies (i.e. TgC y⁻¹).

The experimental NO range is quite low, e.g. NO in global megacities impacted by significant isoprene [NO] range from 1011-1012 molecules cm^{-3} . Yields of first-generation products of the OH-initiated oxidation of isoprene are particularly sensitive in the 108-1012 molecules cm^{-3} range of NO. It would also be good to show how the yields of important detected species in these experiments also change with [NO] (measured and modelled).

Eq 11 and 12 and in text - should organic nitrates be written more logically as RONO2 rather than RO2NO?

Figure 8 – x axis is in "t y⁻¹", please define.

Line 417 – the use of the word “flagrant” is too strong here. The results are contradictory but not in an immoral or obviously offensive way!

Line 433 and 437 – “OH + isoprene representing a next HOM source” and “a next example” what do you mean by “next”?

Version 1:

Reviewer comments:

Reviewer #1

(Remarks to the Author)

The authors have addressed my points. I recommend publication.

Reviewer #2

(Remarks to the Author)

The authors have now addressed the comments that I had for their manuscript. Regarding the atmospheric implication question raised by the other reviewers, I think they have now shown results which show that the levels of isoprene derived HOM-RO₂ comparable with the alpha-pinene derived HOM. As such I would like to recommend this manuscript for publication in Nature Communication.

Reviewer #3

(Remarks to the Author)

I am happy with the authors responses to my reviewer comments, especially the inclusion of a nice comparison with HOM-RO₂ derived from the atmospheric oxidation of a-pinene, giving a better indicating of the relative potential (global) impacts of isoprene photooxidation on biogenically derived HOM species, and hence SOA formation.

I would however like to see the addition of some discussion of a comparison of the relative isoprene and a-pinene derived HOM-RO₂ molar yields in the main text (expanding briefly upon a visual inspection of Fig. 8 vs. Supplementary Fig. S17). This brief discussion could be along the lines of what is already discussed in the main text with respect to the resulting molar HOM-RO₂ production yields shown in Figure 8. However do the a-pinene HOM yields compare to the literature yields given in Bianchi et al., 2019? (reference 24 in the main text).

When introducing the HOM-RO₂ species C₁₀H₁₅O₈, C₁₀H₁₅O₁₀ and C₁₀H₁₇O₇ please reference the appropriate papers that describe the formation mechanisms and structures of these species.

Finally, in Line 424, please alter the sentence slightly to say "Thus, as the next step, the further reactions of ..."

After these minor comments are address, I am happy for this work to be published in Nature Communications.

A point-by-point response to the comments on the manuscript:
Nature Communications NCOMMS-24-54457

Highly oxidized products from the reaction of OH radicals with isoprene under atmospherically relevant NO concentrations

Torsten Berndt^{1*}, Erik H. Hoffmann¹, Andreas Tilgner¹ & Hartmut Herrmann¹

¹ Atmospheric Chemistry Department (ACD), Leibniz Institute for Tropospheric Research (TROPOS), 04318 Leipzig, Germany.

General response to all reviewers:

We thank the three reviewers for their positive and constructive comments and their efforts to improve our manuscript. Below, we provide a point-by-point response to all comments. Comments by the reviewers are given in black normal font, our response to the comments is shown in blue and the changes made in the manuscript and Supplementary Information in red. Please see the manuscript with tracked changes too.

REVIEWER COMMENTS

Reviewer #1 (Remarks to the Author):

The study by Berndt et al. focuses on the investigation of the oxidation of isoprene by OH radicals at atmospheric like conditions with experiments carried on in a laminar flow tube and by measuring oxidation products with a chemical ionization – atmospheric pressure interface – time-of-flight (CI-APi-TOF) instrument.

The focus of the study is the quantification of peroxy radicals which are precursors of highly oxidized molecules (HOMs) which contribute to formation of particles. Based on this study, these radicals are formed with a molar yield of about 0.35% at ambient-like concentrations of NO.

As per usual from previous studies from this group, the manuscript is well written and structured and mostly easy to follow. The chemical paths suggested for the observed radicals and closed-shell species are justified by literature and/or represent a reasonable assumption based on similar molecules. I believe in the capability of this group in detecting and separating such reactive radicals so I do not have any reason to doubt the findings.

We thank the reviewer for the positive feedback regarding the reliability of the experimental findings.

What is not so clear to me is what is the high impact of the study to justify publication on Nature Communications. As pointed out by the authors, the yields found in this study are one order of magnitude higher than what previously reported for studies performed at high concentrations which might not be representative of ambient conditions. Still, the yield remains relatively small. I understand that isoprene emissions are very large so such a small yield can still have a large impact but from the current version of the manuscript it is not clear what the impact is. The authors include a model run and give numbers for the production rate of the radicals they identified but what I am missing is a comparison with other known sources of HOMs. It is mentioned in the paper that these new paths and radicals would contribute in addition to the important source represented by monoterpenes, but no comparison is given so it is hard to judge how much of a contribution this is. Are the authors suggesting that they should be included in global models to bring measurements and modelled results into agreement? It could also be that just the fact that they are able to detect and identify these radicals is per se an excellent achievement so I would suggest stressing this more in the manuscript.

My major point is therefore to clearly show how much this additional source of HOMs contributes to the current known sources. My recommendation would be to run the global

model with the current known sources and add the new yield for isoprene found in this study and see what number (percentage ideally) comes out.

The primary intention was to show a more complete product distribution for atmospherically relevant reaction conditions including the NO concentrations as well as to provide a better mechanistic understanding of the processes going on in the important OH + isoprene reaction in the atmosphere. Special attention was paid to the processes of the δ -RO₂ + NO product channel, hardly taken into account so far, as well as to the HOM formation from the 1,6 H-shift and the δ -RO₂ + NO channel.

But yes, we agree, more information is needed in order to show the significance of isoprene-derived HOM formation within the atmospheric oxidation system, as mentioned by two reviewers.

Hence, we added a comparative modeling study describing the HOM-RO₂ production from the ozonolysis and OH radical reaction of α -pinene. Currently, HOM formation from the oxidation of α -pinene, mainly from the ozonolysis, is considered as an important HOM-generating process. The comparison convincingly confirmed the significance of HOM formation from isoprene, discovering very similar annual HOM-RO₂ production rates from isoprene and α -pinene on global scale. Less oxidized products, other than HOMs, are not incorporated in the comparison as well as potential SOA precursors arising from secondary chemistry, such as IEPOX from OH + HO-C₅H₈OOH (ISOPPOOH) etc. These topics should be subject of a subsequent modeling work and would go beyond the scope of this mainly experimental work.

Changes in the Abstract:

“The isoprene-derived HOM-RO₂ production is comparable with that of α -pinene, currently regarded as very important HOM source.”

Changes in the main text:

“ The isoprene-derived HOM-RO₂ formation was checked against the corresponding processes starting from the ozonolysis and OH radical reaction of α -pinene, which amounts to about one-third of total monoterpene emission.² The oxidation of α -pinene is currently considered as a very important HOM-generating process in the atmosphere.²⁴ Needed modeling calculations were carried out for the year 2010 in an analogous way as described before for OH + isoprene using ECHAM-HAMMOZ including the emission module MEGAN.^{37,38} C₁₀H₁₅O₈ and C₁₀H₁₅O₁₀ (in the following summarized as C₁₀H₁₅O_{8,10}) were considered as the important HOM-RO₂ radicals from α -pinene ozonolysis and C₁₀H₁₇O₇ from the OH radical reaction. NO-dependent molar production yields of C₁₀H₁₅O_{8,10} and C₁₀H₁₇O₇ were determined as explained for the isoprene-related HOM-RO₂ radicals using experimental data as recently reported (Supplementary Fig. S17).^{33,34} The modeling results show an annual production of 3.6×10^6 metric tons of α -pinene-derived HOM-RO₂ radicals, 2.0×10^6 metric tons of C₁₀H₁₅O_{8,10} and 1.6×10^6 metric tons of C₁₀H₁₇O₇, very similar to the total HOM-RO₂ production of 4.3×10^6 metric tons from OH + isoprene (Table 1). Spatial distribution of the annual production of C₁₀H₁₅O_{8,10} and C₁₀H₁₇O₇ is shown in Supplementary Fig. 18 and the fraction of their subsequent reactions either with HO₂ or NO in Table 1.

Table 1: Global HOM-RO₂ production and their fraction reacting with HO₂ and NO.

HOM-RO ₂ radicals	annual production (10 ⁶ metric tons)	reacting with HO ₂ (%)	reacting with NO (%)
C ₅ H ₉ O ₈	0.5	39	61
C ₅ H ₉ O ₉	3.8	54	46
C ₁₀ H ₁₅ O _{8,10}	2.0	44	56
C ₁₀ H ₁₇ O ₇	1.6	23	77

The similar HOM-RO₂ radical production starting from isoprene and α -pinene on global scale is caused by the huge isoprene emission, about 20-fold that of α -pinene,² which compensates the much lower HOM-RO₂ formation yields for isoprene with about only one twentieth compared to the one of α -pinene, cf., Fig. 8 and Supplementary Fig. S17. It should be mentioned at this point, that HOM formation in the isoprene system can also arise from secondary chemistry of isoprene products, i.e., from the reaction of OH radicals with hydroxy hydroperoxides C₅H₁₀O₃ forming especially dihydroxy dihydroperoxides C₅H₁₂O₆.^{4,29,43} The formation pathway leading to C₅H₁₂O₆ and its importance in the atmosphere, however, are currently controversially discussed.^{44,45}

Changes in the SI:

Supplementary Fig. 17: Molar production yields of the HOM-RO₂ radicals C₁₀H₁₅O_{8,10} and C₁₀H₁₇O₇ depending on NO concentration. HOM-RO₂ measurements for the α -pinene ozonolysis were taken from the experiments shown in Figure 5 in reference¹ considering both C₁₀H₁₅O₈ and C₁₀H₁₅O₁₀ in total, here marked as C₁₀H₁₅O_{8,10}. The production of C₁₀H₁₅O_{8,10} was calculated according to equation (SV) in reference¹, identically equal to equation (10) of

the present study, using $k(\text{RO}_2+\text{NO}) = 8.8 \times 10^{-12} \text{ cm}^3 \text{ molecule}^{-1} \text{ s}^{-1}$, $T = 295 \text{ K}$. For $\text{OH} + \alpha$ -pinene, the needed HOM-RO₂ measurements were taken from the experiments depicted in Figure 2a in reference². The corresponding C₁₀H₁₇O₇ production was calculated according to equation (SXII) in reference², again identically equal to equation (10) of the present study, using also $k(\text{RO}_2+\text{NO})$ as mentioned before. In the normalization with respect to converted α -pinene, non-converted initially formed RO₂ radicals, i.e., C₁₀H₁₅O₄ from the ozonolysis¹ and C₁₀H₁₇O₃ from the OH radical reaction², have been considered. Error bars represent the uncertainty of a factor of ~ 2 in the measured HOM-RO₂ concentrations. The measurement point at $[\text{NO}] = 2 \times 10^8 \text{ molecules cm}^{-3}$ in the case of α -pinene ozonolysis represents the production yield for background NO conditions. The dashed lines show assumed extrapolations. Stated average yields for given NO ranges were used in the modeling.

figure 18 not shown

Supplementary Fig. 18: Spatial distribution of HOM-RO₂ radical production from O₃/OH + α -pinene. Annual total production in each grid cell for **A** C₁₀H₁₅O_{8,10} from the ozonolysis of α -pinene and **B** C₁₀H₁₇O₇ from OH + α -pinene.

Minor comment:

Title is a bit weird as with higher I would expect a comparison to show up...higher than previously expected ... for example. As it is now, I am left wondering higher than what?

Thank you for this hint. Obviously, “higher” could probably lead to misunderstanding what is meant. So, we simply changed “higher oxidized” to “highly oxidized” in order to prevent any vagueness.

Change in the title:

“Highly oxidized products from the reaction of OH radicals with isoprene under atmospherically relevant NO concentrations”

Reviewer #2 (Remarks to the Author):

This study by Berndt and colleagues present study on the yield of HOM molecules produced from OH initiated oxidation of isoprene under atmospherically relevant conditions of concentrations for RO₂ and NO. This study pinpoints the structure of molecules that would be dominant for HOMs and the mechanisms for production of such molecules particularly C₅H₉O₈ and C₅H₉O₉. Compared to previous study on a similar topic by Wei Nei et al Nat. Comm. 14, 3347, 2023, the mechanistic study presented here is significantly more focused and robust. Considering that isoprene chemistry plays such a large role in the atmosphere, I would support publication of this study in Nature Communications after the authors have responded to the following comments.

We thank the reviewer for the positive feedback and the recommendation to consider our manuscript for publication in Nature Communications after addressing the comments.

Major comments

-C₅H₈O₂ yield is comparable to the analogue HPALD C₅H₈O₃ even under low NO conditions which seems a bit high. Can the author comment on how fast the alkoxy 1,5 H shift would have to be to account for this? Or are there any other RO₂ in the system such as CH₃O₂ which can form alkoxy in their apparatus?

Yes, the measured C₅H₈O₂ yield was comparable to that of the HPALD, C₅H₈O₃, for low-NO and especially for background NO conditions, e.g., Fig. 4 in the paragraph **Detectable products for background NO level**. The chemical nature of this C₅H₈O₂ and possible formation pathways are not clear at the moment, as already stated in this paragraph. Assuming that C₅H₈O₂ represents the unsaturated hydroxy carbonyl **24** (Fig. 3), it should be formed via the 1,5 H-shift of **9** forming **11**. This isomerization step is super fast according to Dibble, reference 11, meaning that all alkoxy **9** is immediately converted to **11**. The bottleneck here seems to be the production of the alkoxy **9**. The reaction of background NO with HO-C₅H₈O₂ cannot explain the production of the alkoxy **9** needed to describe the measured C₅H₈O₂ production, as stated in the manuscript. If any HO-C₅H₈O₂ self- and cross-reactions with other RO₂ (including acetyl peroxy radicals from TME ozonolysis or any background RO₂ such as CH₃O₂) would significantly form alkoxy **9**, a more than linear increase of C₅H₈O₂ with rising isoprene conversion should be visible due to the 2nd order kinetics of the bimolecular HO-C₅H₈O₂ reactions. However, this is not the case as clearly seen also in additional experiments, not mentioned in the manuscript. The fact that RO₂ + RO₂ is most likely not the source for C₅H₈O₂ under low-NO conditions is stated now in the manuscript. So, the source of C₅H₈O₂ for low-NO level remains undiscovered at the moment. On the other hand, even small NO additions lead to a significant increase of the C₅H₈O₂ signal, see Supplementary Fig. 4, here most likely due to the production of the unsaturated hydroxy carbonyl **24**, well in line with the expectation based on the mechanism.

Changes in the main text:

“Moreover, significant contributions from HO-C₅H₈O₂ radical self- and cross-reactions are unlikely, because of the 2nd order kinetics of this process, which is contrary to the observed linear increase of C₅H₈O₂ with rising isoprene conversion as for the 1,6 H-shift products.”

-Medeiros and co-authors (JACS Au 2022, 2, 809-818) have recently suggested that there is a significant OH production from dihydrocarbonyl pathways which would have been verifiable from this study. However, the yield of the co-C₄H₈O₅ product in that pathway is measured to

be low here and seems to be in contradiction. The authors should comment on this.

Medeiros et al. (2022) proposed OH and C₄H₈O₅ **20** formation from the dihydroperoxy carbonyl **18** as the predominant process of the 1,6 H-shift channel for atmospheric conditions based on their high-temperature OH recycling experiments. This result is in clear contradiction to the findings of the present work and other investigations. This is stated now in the manuscript according to the reviewer's suggestion.

Changes in the main text:

“Furthermore, the formation of C₄H₈O₅ **20**, the expected decomposition product of C₅H₉O₅ radicals **18** (Fig. 2), was only a minor process (Fig. 4) being in line with our former findings.¹⁴ This is in contradiction to conclusions based on high-temperature OH radical recycling experiments (480 - 584 K) stating the formation and subsequent decomposition of C₅H₉O₅ **18** as the main process of 1,6 H-shift product formation for atmospheric conditions.³¹”

-Is there any reason why NO cannot be quantified below 2×10^8 molecule cm⁻³ concentration?

A measurement device for such small NO concentrations is (at least) commercially not yet available. Thus, we used the measured onset of organic nitrate detection for NO additions of about 3×10^8 molecules cm⁻³ in order to estimate the stated background NO level of $< 2 \times 10^8$ molecules cm⁻³, see Supplementary Fig. 4.

Minor comments

Bring Figure S1 to the main text

The former Supplementary Fig. 1 has been moved to the main text, now it is Fig. 2.

Changes in the main text:

Fig. 2: Proposed reaction scheme for consecutive reactions of the allyl radical 10 from the 1,6 H-shift channel. The scheme is consistent with the observed product formation from the present study as well as partly with data from the literature.^{3-5,7,14,16} Signals of detected RO₂ radicals are highlighted in yellow and those from closed-shell products in blue. Only important main products of individual pathways are shown. Bimolecular reactions of RO₂ radicals with NO and HO₂ radicals are not depicted.

Line 34, change ‘emphasis’ to ‘emphasize’

A correction was done.

Change in the text:

“Overall, this study provides a more complete insight into isoprene’s degradation process and emphasizes the importance of HOM formation.”

Line 42, change ‘leads to a variety’ to ‘leads to production of a variety’

The correction was done.

Change in the text:

“Its dominant atmospheric sink is the gas-phase reaction with OH radicals, which leads to production of a variety of RO₂ radicals due to isoprene’s conjugated diene structure.”

Line 68, change ‘Predominate’ to ‘The main products’

The correction was done.

Change in the text:

“The main products of the HO-C₅H₈O₂ radical reactions with HO₂ are the corresponding hydroxy hydroperoxides, ROOHs.^{4,9}”

Line 84, delete ‘also’ change ‘as well as similar’ to ‘and’

The correction was done.

Changes in the text:

“Thus, a similar chemical behavior and similar product channels can be expected in their further reactions.”

Line 112, reword to ‘allyl radical 11 can be produced via the alkoxy channels of the reactions between 4 or 6 with other RO₂ radicals’

The correction was done.

Changes in the text:

“A signal consistent with the formation of the highest oxidized RO₂ radical **27** was already observed under conditions of significant RO₂ self- and cross-reactions in experiments where the allyl radical **11** can be produced via the alkoxy channels of the reactions between **4** or **6** with other RO₂ radicals.^{16,22}”

Line 125, delete ‘at the minimum’

The correction was done.

Changes in the text:

“The experiments were conducted at 295 ± 0.5 K under atmospheric conditions in a laminar flow tube (LFT)^{29,30} with a reaction time of 32 s, thus, coming close to conditions of the equilibrated initial RO₂ radical distribution as existing in the atmosphere.”

Line 131, change ‘being’ to ‘which are’

The correction was done.

Changes in the text:

“NO concentrations ranged from $< 2 \times 10^8$ to 8.3×10^{10} molecules cm^{-3} which are relevant for remote to urban areas.”

Line 193, change ‘substance flow’ to ‘reaction flux’

‘substance flow’ was changed to ‘reaction flux’ throughout in the text and the SI.

Changes in the text:

“Using TME ozonolysis for OH generation, the reaction flux over the 1,6 H-shift channel initially declined with rising NO, but increased afterwards due to rising importance of the $\text{NO} + \text{HO}_2 \rightarrow \text{OH} + \text{NO}_2$ reaction leading to rising isoprene conversion, see results from modeling in Supplementary Fig. 5.”

And at other places in the text and SI.

Line 253-54, this sentence needs better wording

This sentence has been reworded. Now it should be easier to understand.

Changes in the text:

“Formation yields with respect to converted isoprene of 19.3 ± 6.1 and 3.3 ± 1.6 % for $\text{C}_5\text{H}_8\text{O}_2$ and $\text{C}_4\text{H}_6\text{O}_2$, respectively,²⁶ were found in a flow tube study using elevated concentration levels, which stands for higher $\delta\text{-RO}_2 + \text{NO}$ channel-specific yields of both products than observed in the present study (~ 100 % in the case of $\text{C}_5\text{H}_8\text{O}_2$).”

Line 258, change ‘definitionally’ to ‘by definition’

The correction was done.

Changes in the text:

“Nitrate ionization for product analysis was chosen in the experiments for selective and sensitive detection of HOMs, which should contain by definition at least 6 O atoms in the molecule.²⁴”

Line 359, check ‘17+/-4’ to ‘19+/-3’

Thank you very much for careful reading of the manuscript, the two numbers were swapped.

Changes in the text:

“Thus, in the atmosphere, total closed-shell HOM yields of 39 ± 8 % and 19 ± 3 % can be expected from the reaction of $\text{C}_5\text{H}_9\text{O}_8$ and $\text{C}_5\text{H}_9\text{O}_9$ with NO, respectively.”

Line 404, delete ‘finally important for SOA generation’

The correction was done.

Changes in the text:

“This means that the closed-shell HOM formation is governed by the HOM-RO₂ + HO₂ reaction.”

Line 479, reword

This sentence has been reworded. The new two sentences should be easier to understand.

Changes in the text:

“As the reagent ions in the product measurements served ethylammonium (C₂H₅NH₃⁺), iodide (I⁻) and nitrate (NO₃⁻). Ammonium (NH₄⁺) was used in additional calibration experiments.”

Line 568, its not obvious where 0.5 comes from

It simply comes from the integral $\int (a \times t) dt = 0.5 \times a \times t^2$. Now it is shown more explicitly where the final relationship, now equation (12), comes from.

Changes in the text:

$$[\text{HOM} - \text{RO}_2] = f(t) = a \times t, \quad a = \text{constant} \quad (8)$$

Insertion of equation (8) into equation (7) yields:

$$\frac{d[\text{HOM} - \text{RO}_2]}{dt} = P_{\text{HOM-RO}_2} - k_{\text{RO}_2+\text{NO}} \times [\text{NO}] \times a \times t \quad (9)$$

Integration of equation (9) with $[\text{HOM} - \text{RO}_2] = 0$ at $t = 0$ and $[\text{NO}]_t = [\text{NO}]_0$ leads to:

$$[\text{HOM} - \text{RO}_2]_t = \int_0^t P_{\text{HOM-RO}_2} dt - 0.5 k_{\text{RO}_2+\text{NO}} \times [\text{NO}]_0 \times a \times t^2 \quad (10)$$

Make use of equation (8) in equation (10), it follows:

$$[\text{HOM} - \text{RO}_2]_t = \int_0^t P_{\text{HOM-RO}_2} dt - 0.5 k_{\text{RO}_2+\text{NO}} \times [\text{NO}]_0 \times [\text{HOM} - \text{RO}_2]_t \times t \quad (11)$$

Thus, the integral HOM-RO₂ formation as a function of NO for a given time t can be given as:

$$\int_0^t P_{\text{HOM-RO}_2} dt = [\text{HOM} - \text{RO}_2]_t \times (1 + 0.5 k_{\text{RO}_2+\text{NO}} \times [\text{NO}]_0 \times t) \quad (12)$$

Reviewer #3 (Remarks to the Author):

Berndt et al. present a very nice detailed mechanistic study on the formation of a complex range of OH + isoprene products (some for the first time) that can lead to highly oxygenated molecular precursors to secondary organic aerosol, which have been assigned to different RO₂ production channels, one which is prevalent under pristine atmospheric conditions and one which is potentially active under moderate NO conditions (at least as shown under the conditions of the experiments). The paper is well written and the experimental approach is reasonably sound given the complexities of the system being studied and the low concentrations of the species under investigation. There is a large amount of uncertainties in these experiments, as highlighted by the authors, leading to some speculation in the impacts of the results.

We thank the reviewer for the very positive feedback regarding the experimental output and for the manuscript itself.

It is the potential importance of these results on global atmospheric chemistry which is not what I am really getting from the way they are presented here. How does the potential to impact biogenic SOA formation compare to what the regional and global models already predict coming from other, more high yielding isoprene product precursor species such as, for example, isoprene epoxides (IEPOX) (low NO pathways) or e.g. 2-methylglyceric acid (high NO pathways)? How do these new potential SOA formation pathways compare to those from monoterpenes (these routes briefly mentioned in the paper)? For this to be a Nature Communications paper, I would like to see more discussions on the potential regional and global impacts of these results. Having said that, as the paper is written, it would make a solid publication in a more atmospheric chemistry focused journal as the results will be very interesting to atmospheric chemical mechanism developers and modellers.

The most critical point addressed by reviewer 3, i.e., the impact of isoprene-derived HOMs, is identical to that stated by reviewer 1. So, we want to answer just as before:

The primary intention was to show a more complete product distribution for atmospherically relevant reaction conditions including the NO concentrations as well as to provide a better mechanistic understanding of the processes going on in the important OH + isoprene reaction in the atmosphere. Special attention was paid to the processes of the δ -RO₂ + NO product channel, hardly taken into account so far, as well as to the HOM formation from the 1,6 H-shift and the δ -RO₂ + NO channel.

But yes, we agree, more information is needed in order to show the significance of isoprene-derived HOM formation within the atmospheric oxidation system, as mentioned by two reviewers.

Hence, we added a comparative modeling study describing the HOM-RO₂ production from the ozonolysis and OH radical reaction of α -pinene. Currently, HOM formation from the oxidation of α -pinene, mainly from the ozonolysis, is considered as an important HOM-generating process. The comparison convincingly confirmed the significance of HOM formation from isoprene, discovering very similar annual HOM-RO₂ production rates from isoprene and α -pinene on global scale. Less oxidized products, other than HOMs, were not incorporated in the comparison as well as potential SOA precursors arising from secondary chemistry, such as IEPOX from OH + HO-C₅H₈OOH (ISOPPOOH) etc. These topics should be subject of a subsequent modeling work and would go beyond the scope of this mainly experimental work.

Changes in the Abstract:

“The isoprene-derived HOM-RO₂ production is comparable with that of α -pinene, currently regarded as very important HOM source.”

Changes in the main text:

“ The isoprene-derived HOM-RO₂ formation was checked against the corresponding processes starting from the ozonolysis and OH radical reaction of α -pinene, which amounts to about one-third of total monoterpene emission.² The oxidation of α -pinene is currently considered as a very important HOM-generating process in the atmosphere.²⁴ Needed modeling calculations were carried out for the year 2010 in an analogous way as described before for OH + isoprene using ECHAM-HAMMOZ including the emission module MEGAN.^{37,38} C₁₀H₁₅O₈ and C₁₀H₁₅O₁₀ (in the following summarized as C₁₀H₁₅O_{8,10}) were considered as the important HOM-RO₂ radicals from α -pinene ozonolysis and C₁₀H₁₇O₇ from the OH radical reaction. NO-dependent molar production yields of C₁₀H₁₅O_{8,10} and C₁₀H₁₇O₇ were determined as explained for the isoprene-related HOM-RO₂ radicals using experimental data as recently reported (Supplementary Fig. S17).^{33,34} The modeling results show an annual production of 3.6×10^6 metric tons of α -pinene-derived HOM-RO₂ radicals, 2.0×10^6 metric tons of C₁₀H₁₅O_{8,10} and 1.6×10^6 metric tons of C₁₀H₁₇O₇, very similar to the total HOM-RO₂ production of 4.3×10^6 metric tons from OH + isoprene (Table 1). Spatial distribution of the annual production of C₁₀H₁₅O_{8,10} and C₁₀H₁₇O₇ is shown in Supplementary Fig. 18 and the fraction of their subsequent reactions either with HO₂ or NO in Table 1.

Table 1: Global HOM-RO₂ production and their fraction reacting with HO₂ and NO.

HOM-RO ₂ radicals	annual production (10 ⁶ metric tons)	reacting with HO ₂ (%)	reacting with NO (%)
C ₅ H ₉ O ₈	0.5	39	61
C ₅ H ₉ O ₉	3.8	54	46
C ₁₀ H ₁₅ O _{8,10}	2.0	44	56
C ₁₀ H ₁₇ O ₇	1.6	23	77

The similar HOM-RO₂ radical production starting from isoprene and α -pinene on global scale is caused by the huge isoprene emission, about 20-fold that of α -pinene,² which compensates the much lower HOM-RO₂ formation yields for isoprene with about only one twentieth compared to the one of α -pinene, cf., Fig. 8 and Supplementary Fig. S17. It should be mentioned at this point, that HOM formation in the isoprene system can also arise from secondary chemistry of isoprene products, i.e., from the reaction of OH radicals with hydroxy hydroperoxides C₅H₁₀O₃ forming especially dihydroxy dihydroperoxides C₅H₁₂O₆.^{4,29,43} The formation pathway leading to C₅H₁₂O₆ and its importance in the atmosphere, however, are currently controversially discussed.^{44,45}

Changes in the SI:

Supplementary Fig. 17: Molar production yields of the HOM-RO₂ radicals C₁₀H₁₅O_{8,10} and C₁₀H₁₇O₇ depending on NO concentration. HOM-RO₂ measurements for the α -pinene ozonolysis were taken from the experiments shown in Figure 5 in reference¹ considering both C₁₀H₁₅O₈ and C₁₀H₁₅O₁₀ in total, here marked as C₁₀H₁₅O_{8,10}. The production of C₁₀H₁₅O_{8,10} was calculated according to equation (SV) in reference¹, identically equal to equation (10) of the present study, using $k(\text{RO}_2+\text{NO}) = 8.8 \times 10^{-12} \text{ cm}^3 \text{ molecule}^{-1} \text{ s}^{-1}$, $T = 295 \text{ K}$. For OH + α -pinene, the needed HOM-RO₂ measurements were taken from the experiments depicted in Figure 2a in reference². The corresponding C₁₀H₁₇O₇ production was calculated according to equation (SXII) in reference², again identically equal to equation (10) of the present study, using also $k(\text{RO}_2+\text{NO})$ as mentioned before. In the normalization with respect to converted α -pinene, non-converted initially formed RO₂ radicals, i.e., C₁₀H₁₅O₄ from the ozonolysis¹ and C₁₀H₁₇O₃ from the OH radical reaction², have been considered. Error bars represent the uncertainty of a factor of ~ 2 in the measured HOM-RO₂ concentrations. The measurement point at $[\text{NO}] = 2 \times 10^8 \text{ molecules cm}^{-3}$ in the case of α -pinene ozonolysis represents the production yield for background NO conditions. The dashed lines show assumed extrapolations. Stated average yields for given NO ranges were used in the modeling.

figure 18 not shown

Supplementary Fig. 18: Spatial distribution of HOM-RO₂ radical production from O₃/OH + α -pinene. Annual total production in each grid cell for **A** C₁₀H₁₅O_{8,10} from the ozonolysis of α -pinene and **B** C₁₀H₁₇O₇ from OH + α -pinene.

Some more specific point are given below:

The mechanistic background to this work is mainly focused around that of the Wennberg et al., 2018 review, and some subsequent studies after this. However, how do the pathways presented here compare to those given in other detailed isoprene mechanisms used extensively in atmospheric models, such as the MCMv3.3.1 isoprene scheme (Jenkin et al., 2015 (www.atmos-chem-phys.net/15/11433/2015/)) and the new Jülich isoprene mechanisms (e.g. Tsiligiannis et al., 2022; <https://doi.org/10.1029/2021gl097366>), whose initial OH-isoprene adducts with O₂ chemistry seems more complex than that presented here?

The basic mechanistic understanding regarding the first steps of the OH + isoprene reaction for atmospheric conditions is mainly due to the pioneering work by J. Peeters et al., the Leuven group, as cited in the manuscript in refs. 3 and 7. After that, in 2017 Wennberg et al., the Caltech group, published results from an impressive experimental study on the “RO₂ dynamics” improving the quality of the kinetic data especially for the primary RO₂ chemistry in this system, refs. 4 and 5 in the manuscript. Both the Leuven and the Caltech mechanism are focusing on the terminal addition of OH and the subsequent RO₂ chemistry. The MCMv3.3.1 mechanism by Jenkin et al., ACP_2015, as mentioned by the reviewer, also considers the OH addition in non-terminal position. But, these steps only amount to about 8% of the initial OH attack, not significantly change the subsequent product formation. Novelli et al., ref. 21 in the manuscript, the Jülich group, used the mechanisms from Leuven, Caltech and MCM with some modifications for the following-up chemistry of C₅H₉O₅ radicals **18** in order to compare chamber results with modeling output. The mentioned Jülich mechanism by Tsiligiannis et al., GRL_2022, describes the NO₃ chemistry.

All in all, the used mechanism in the present work based on the initial RO₂ chemistry given in the Leuven and Caltech mechanism along with the kinetic data provided by Caltech seems to be the most reliable approach at the moment.

Changes in the text:

“Results from modeling of the initial RO₂ radical processes^{4,7} using latest kinetic data^{4,5} allowed to assess the importance of the 1,6 H-shift and the δ -RO₂ + NO channel for product formation in the experiments, see Supplementary Table 1.”

MCMv3.3.1 website is referenced (24). It would be better to reference the paper that is focused on the MCMv3.3.1 isoprene mechanism update instead, i.e. Jenkin et al., 2015 (www.atmos-chem-phys.net/15/11433/2015/).

Thank you for this hint. The reference is given now in the form as requested at the MCM website.

Changed reference:

“The chemical mechanistic information was taken from the Master Chemical Mechanism, MCM v3.3.1 (Jenkin et al., *Atmos. Chem. Phys.*, **15**, 11433-11459, (2015)), via website: www.mcm.york.ac.uk. “

Throughout the text the emissions are given in metric tonnes per year. Please convert to more conventional units in order to compare to other global modelling studies (i.e. TgC y⁻¹).

We used for the absolute mass (not only for the carbon) the unit “metric tons“ or mostly “ 10^6 metric tons“, which is identical with “Tg”, to make this huge amount better understandable for a wide readership of Nat.Comm., not only for experts in global modeling. So, we would like keep this unit throughout in the text and figures.

The experimental NO range is quite low, e.g. NO in global megacities impacted by significant isoprene [NO] range from 1011-1012 molecules cm^{-3} . Yields of first-generation products of the OH-initiated oxidation of isoprene are particularly sensitive in the 108-1012 molecules cm^{-3} range of NO. It would also be good to show how the yields of important detected species in these experiments also change with [NO] (measured and modelled).

The NO concentrations in the experiments ranged from $< 2 \times 10^8$ to 8.3×10^{10} molecules cm^{-3} which are relevant for remote to urban area. For the highest NO, a bimolecular RO_2 reactivity of 0.7 s^{-1} follows, at which the fate of initially formed RO_2 radicals becomes less dependent on further NO increase, see Fig. 4 in P. Wennbergs review, ref. 4 in the manuscript. Thus, we think that an appropriate NO range was considered in the experiments in order to investigate the important processes of initially formed RO_2 radicals.

Yes, the reviewer is right. A comparison “experiment vs. model output” helps to assess how good we understand a reaction system. The applied modeling focuses on the initial RO_2 radical processes incl. the reaction flux over the 1,6 H-shift and the $\delta\text{-RO}_2 + \text{NO}$ channel, not describing the formation of 1st generation products, such as $\text{C}_5\text{H}_8\text{O}_2$, $\text{C}_5\text{H}_8\text{O}_3$ etc., in detail. A comparison of the reaction flux over both channels from modeling with experimental product data, Supplementary Fig. 5 vs. Fig. 5 and Supplementary Fig. 7 vs. Supplementary Fig. 8, shows that the individual product concentrations qualitatively behave like the reaction flux over the corresponding channel with rising NO. A more quantitative analysis seems to be impossible at the moment.

Eq 11 and 12 and in text - should organic nitrates be written more logically as RONO_2 rather than RO_2NO ?

It is right, RONO_2 is logically better than RO_2NO . Thus, this was changed throughout in the text and in the figures.

Figure 8 – x axis is in “t y-1”, please define.

Now the x-axis label is given as “metric tons per year”, should be easier to understand.

Line 417 – the use of the word “flagrant” is too strong here. The results are contradictory but not in an immoral or obviously offensive way!

This sentence has been reworded.

Changes in the text:

“On the other hand, a HOM yield of up to 11% as proposed from theoretical calculations for low-NO conditions¹⁶ appears to be definitely too high compared to the experimentally based findings.”

Line 433 and 437 – “OH + isoprene representing a next HOM source” and “a next example”
what do you mean by “next”?

This has been reworded.

Changes in the text:

“Although there exists lack of knowledge regarding the product yield from the HOM-RO₂ + HO₂ reaction, annual closed-shell HOM production in the order of a few 10⁶ metric tons globally can be expected from the OH + isoprene reaction representing an additional HOM source to the important one from the oxidation of α-pinene.²⁴”

Other changes in the manuscript

Unfortunately, we found a small bug in the conversion routines regarding the applied RO₂ yields for isoprene oxidation. This has been fixed and new simulations were performed, accordingly. The calculations of the HOM-RO₂ radical production were designed for the newly changed yields. As a result, higher production of HOM-RO₂ radicals is simulated now. In addition, we checked all offline calculations and revised when necessary. Here, the previous calculation on the further reaction of the HOM-RO₂ radical with either HO₂ or NO were reinvestigated. The current offline approach has resulted into an overestimation of the yields, not in-line with the online calculated ones for isoprene-related RO₂ radicals. Thus, the values given in the manuscript were updated, please see the manuscript with tracked changes for details.

A point-by-point response to the comments on the manuscript:
Nature Communications NCOMMS-24-54457A

Highly oxidized products from the atmospheric reaction of hydroxyl radicals with isoprene

Torsten Berndt^{1*}, Erik H. Hoffmann¹, Andreas Tilgner¹ & Hartmut Herrmann¹

¹ Atmospheric Chemistry Department (ACD), Leibniz Institute for Tropospheric Research (TROPOS), 04318 Leipzig, Germany.

General response to all reviewers:

We thank all three reviewers for their very positive feedback. Below, we provide a point-by-point response to the additional comments of reviewer 3. Comments by the reviewer are given in black normal font, our response to the comments is shown in blue and the changes made in the manuscript and Supplementary Information in red.

REVIEWER COMMENTS

Reviewer #3 (Remarks to the Author):

I am happy with the authors responses to my reviewer comments, especially the inclusion of a nice comparison with HOM-RO₂ derived from the atmospheric oxidation of α -pinene, giving a better indicating of the relative potential (global) impacts of isoprene photooxidation on biogenically derived HOM species, and hence SOA formation.

I would however like to see the addition of some discussion of a comparison of the relative isoprene and α -pinene derived HOM-RO₂ molar yields in the main text (expanding briefly upon a visual inspection of Fig. 8 vs. Supplementary Fig. S17). This brief discussion could be along the lines of what is already discussed in the main text with respect to the resulting molar HOM-RO₂ production yields shown in Figure 8. However do the α -pinene HOM yields compare to the literature yields given in Bianchi et al., 2019? (reference 24 in the main text).

A short comparison of HOM-RO₂ yields from isoprene and α -pinene oxidations is given now. We also added a short discussion regarding the HOM yields of α -pinene incl. the insufficient detection sensitivity for HOMs arising from the OH reaction.

Changes in the text:

“For low-NO conditions, molar production yields of $6.9^{+6.9}_{-3.5}\%$ for C₁₀H₁₅O_{8,10} and $7.6^{+7.6}_{-3.8}\%$ for C₁₀H₁₇O₇ were obtained being a factor of about 20 higher compared with the results from isoprene oxidation. The α -pinene-derived molar yields are at the upper end or significantly higher compared with the reported HOM data (mainly for closed-shell HOMs) measured by means of nitrate ionization.²⁴ It is to be noted here, that the nitrate ionization scheme appears to be less sensitive especially for C₁₀H₁₇O₇ and subsequent products.^{33,47”}

When introducing the HOM-RO₂ species C₁₀H₁₅O₈, C₁₀H₁₅O₁₀ and C₁₀H₁₇O₇ please reference the appropriate papers that describe the formation mechanisms and structures of these species.

We added a series of references describing the formation of these HOM-RO₂ radicals and showing proposed structures. 8 new references are included now, refs 43 - 50.

Changes in the text:

“C₁₀H₁₅O₈ and C₁₀H₁₅O₁₀ (in the following summarized as C₁₀H₁₅O_{8,10}) were considered as the important HOM-RO₂ radicals from α-pinene ozonolysis^{22,34,43-46} and C₁₀H₁₇O₇ from the OH radical reaction^{33,47-50}.”

Finally, in Line 424, please alter the sentence slightly to say "Thus, as the next step, the further reactions of ..."

We followed reviewer's suggestions and changed the sentence accordingly.

Changes in the text:

“Thus, as the next step, the further reactions of C₅H₉O₈ and C₅H₉O₉ radicals with NO and HO₂ were considered in the global modeling.”